# Spatial Memory for Out-of-Vision Manipulation in Vision-Language-Action

**Pengteng Li** [1 2]  **Weiyu Guo** [1 2]  **He Zhang** [1 2]  **Tiefu Cai** [1 2]  **Xiao He** [2]  **Yandong Guo** [2]  **Hui Xiong** [1 3]

## Abstract

We introduce **SOMA**, the **S**patial memory framework for **O**ut-of-Vision **M**anipulation in Vision-Language-**A**ction (VLA) models. Most existing VLAs implicitly assume that task-relevant objects are always visible, leading to brittle and reactive behaviors when targets fall outside the camera's field of view. SOMA addresses this limitation by equipping VLAs with a persistent, spatial memory constructed from multi-view observations acquired via a movable head camera, enabling reasoning beyond the current visual frustum. The framework consists of three components: *Spatial Memory Construction* for aggregating angular-wise observations into a unified spatial–semantic representation by scanning, *Dynamic Memory Refinement* for maintaining global consistency over time, and *Contextual Memory Retrieval* for activating instruction-relevant spatial cues during manipulation. We evaluate SOMA on five self-designed challenging real-world OOV manipulation tasks, including multi-step and dual-arm scenarios, where target objects are initially invisible. Experiment results show that SOMA not only improves task success rates, but also induces qualitatively different manipulation behaviors, with faster target localization, reduced viewpoint search, and near one-shot grasping under partial observability. Additional experiments on RoboCasa GR1 and SimplerEnv further validate the effectiveness of SOMA's memory design under conventional fully observable settings. Code will be released soon.

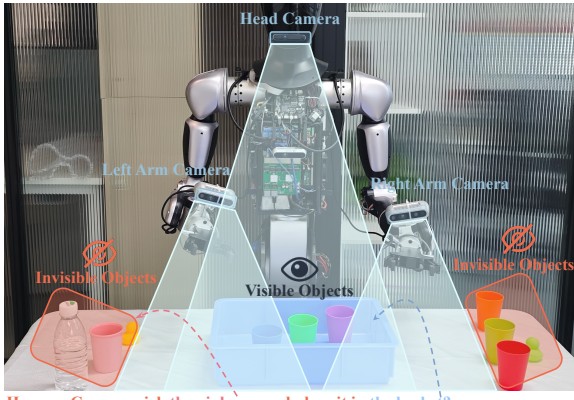

*Figure 1.* Illustration of the Out-of-Vision (OOV) limitation in existing VLA models. Most VLAs rely on purely reactive perception—actions are driven only by what is visible in the current view. When the target moves outside the field of view, perception can no longer support manipulation, leading to inevitable task failure.

## 1. Introduction

The development of VLAs have become a central direction in robotic action modeling research (Zhao et al., 2025; Chen et al., 2025c; Kim et al., 2024; Shukor et al., 2025; Intelligence et al., 2025). These systems typically extend large-scale pre-trained Multimodal Large Language Models (MLLMs) (Bjorck et al., 2025; Yang et al., 2025a) with an action head or specialized action module that maps multimodal inputs—such as visual observations and natural-language instructions—into executable robotic actions. By leveraging the strong perception and reasoning capabilities of MLLMs, VLA models demonstrate improved generalization and task versatility across diverse robotic manipulation scenarios (Jiang et al., 2025b; Bu et al., 2025).

However, most existing VLAs are developed under fixed-view tabletop manipulation setups, typically relying on a single static camera or a third-person viewpoint. Such configurations are widely adopted because they simplify camera–robot calibration, provide stable state estimation, and facilitate large-scale data collection without the need for multi-view coordination or explicit spatial fusion. As a result, these models implicitly operate under a view-bound assumption—namely, that *the object referenced in the instruction is visible within the robot's current camera view at decision time.* This assumption fundamentally limits the

---

[1]Thrust of Artificial Intelligence, The Hong Kong University of Science and Technology (Guangzhou), China [2]AI[2]Robotics [3]Department of Computer Science and Engineering, The Hong Kong University of Science and Technology Hong Kong SAR, China. Correspondence to: Yandong Guo <Yandong.guo@live.com>, Hui Xiong <xionghui@ust.hk>.

*Proceedings of the $43^{rd}$ International Conference on Machine Learning*, Seoul, South Korea. PMLR 306, 2026. Copyright 2026 by the author(s).

model's ability to reason about objects that are temporarily occluded or outside the field of view. Without a mechanism to maintain a persistent spatial representation of the scene, the perception–action loop becomes strictly view-dependent: when a target object is not observed, the model lacks the necessary context to infer its presence or spatial relation, often leading to failure. As shown in Figure 1, even when multiple fixed camera viewpoints (e.g., head- and arm-mounted cameras) are available, the absence of a global spatial memory prevents the system from localizing unseen targets (such as the pink cup), highlighting a core limitation of view-bound perception in current VLA models.

To address this limitation, recent work has attempted to compensate for out-of-view perception by relying on internal spatial reasoning under static camera setups, inferring unseen targets via learned spatial priors (Team et al., 2025; Jiang et al., 2025a; Bu et al., 2025; Li et al., 2025c). However, such inference is inherently brittle: existing MLLMs can reason reliably only when partial visual evidence or object-related cues are available, and their spatial estimates deteriorate rapidly once the target is fully outside the observable field (Yang et al., 2025b). When inferred relations deviate from physical reality, errors propagate through the perception–action pipeline, leading to inaccurate localization, misaligned motion, or task failure. These observations expose a fundamental limitation of current VLA paradigms: spatial reasoning alone, without grounded perceptual evidence, is insufficient for robust manipulation. When objects have never been observed or when past observations are not retained, increasing model capacity or reasoning depth cannot compensate for missing perceptual grounding. *Addressing this gap requires mechanisms that both acquire spatial evidence beyond the current view and retain it in a persistent scene representation.* In particular, integrating angular-wise observations into a coherent spatial–semantic memory enables globally consistent reasoning and effective manipulation even when task-relevant objects are temporarily out of view.

Based on these insights, we introduce **SOMA**, a VLA framework for out-of-vision manipulation that equips the robot with persistent spatial memory for reasoning and action. Unlike prior VLAs that rely solely on instantaneous, view-bound visual observations, SOMA explicitly models perception as a memory-centric process, enabling manipulation beyond the current camera view. SOMA is built upon two key design principles: (1) a movable head-mounted camera that provides access to diverse viewpoints of the workspace, and (2) a global spatial memory that integrates angular-wise observations into a unified, queryable spatial–semantic representation. When a task-relevant target cannot be localized within the current view, SOMA triggers its *Spatial Memory Construction* phase, during which the camera follows a predefined scanning strategy to systemat-

ically acquire multi-view observations and assemble a coherent spatial representation of the scene. This process employs a multi-level visual abstraction pipeline that combines category-level semantics (Cheng et al., 2024), fine-grained appearance priors (Siméoni et al., 2025), and implicit 3D geometric cues (Wang et al., 2025b). By fusing these complementary signals, SOMA constructs an overview memory that serves as a structured spatial substrate for downstream manipulation. During interaction, *Dynamic Memory Refinement* incrementally assimilates newly observed information through similarity-aware fusion, maintaining global consistency as the scene evolves. Meanwhile, *Contextual Memory Retrieval* aligns incoming multimodal instruction and perception tokens with the spatial memory, selectively activating instruction-relevant regions to guide precise and grounded manipulation, even when target objects are temporarily outside the current field of view. Extensive real-world evaluations demonstrate that SOMA fundamentally alters manipulation behavior under OOV conditions, enabling faster target localization, reduced viewpoint search, and near one-shot grasping. Additional experiments on RoboCasa Tabletop GR1 (Nasiriany et al., 2024) and SimplerEnv (Li et al., 2024) further validate the robustness and generality of SOMA's memory mechanism under standard observability settings. Overall, our contribution can be listed as follows:

- We introduce **SOMA**, a spatial memory framework for Vision-Language-Action models that enables out-of-vision manipulation by inducing a memory-guided manipulation paradigm beyond the current camera view, rather than reactive view-bound exploration.

- We propose a memory-centric perception architecture comprising *Spatial Memory Construction*, *Dynamic Memory Refinement*, and *Contextual Memory Retrieval*, which together maintain a persistent, object-centric spatial representation that supports cross-view consistency and instruction-aware action selection under limited viewpoint coverage.

- Real-world experiments show that SOMA not only improves task success rates, but also consistently yields qualitatively different manipulation behaviors under out-of-vision conditions, including earlier target fixation, reduced head exploration, and near one-shot grasping. Additional evaluations on RoboCasa GR1 and SimplerEnv further validate the effectiveness of our memory design under fully observable settings

## 2. Related Work

**Policy Learning for Robotic Manipulation.** Policy learning underpins robotic manipulation, enabling data-driven control for arm and whole-body coordination (Du & Song, 2025; Sun & Song, 2025; He et al., 2025; 2024; Xie et al.,

2025; Weng et al., 2025; Fu et al., 2024; Song et al., 2025a; 2024). Recent work introduces movable cameras to improve exploration under partial observability (Xiong et al., 2025; Zeng et al., 2025), but these methods remain instruction-agnostic and rely on policies (*e.g.*, Diffusion Policy (Chi et al., 2025)) without persistent spatial memory, limiting generalization to unseen goals and spatial configurations. In contrast, our framework integrates movable perception into a VLA models, enabling instruction-aligned, memory-guided manipulation.

**Vision-Language-Action Models.** VLA models provide a unified framework for instruction-driven robotic manipulation. Prior work spans data-driven approaches trained on large-scale multimodal datasets (Luo et al., 2025; Bu et al., 2025; Zhai et al., 2025; Jiang et al., 2025a; Zitkovich et al., 2023; Intelligence et al., 2025; Guo et al., 2026), world-model (Wan et al., 2025; Agarwal et al., 2025; Li et al., 2025d) and latent-action formulations that enhance generalization (Cen et al., 2025; Zhang et al., 2025a; Agarwal et al., 2025; Ye et al., 2024; Chen et al., 2025d), and extensions to diverse embodiments including mobile, tactile, and force-based platforms (Chen et al., 2025b; Zhang et al., 2025c; Cheng et al., 2025; Yu et al., 2025). Recent efforts also explore efficiency (Chen et al., 2025a; Wang et al., 2025a; Zhang et al., 2025b; Wang et al., 2025d; Liu et al., 2024; Song et al., 2025b; Zheng et al., 2025; Wang et al., 2025c) and spatial grounding via geometric cues for precise localization (Qu et al., 2025; Li et al., 2025a; Zhu et al., 2025; Yuan et al., 2025; Li et al., 2025b; Guo et al., 2022; Li et al., 2025e; Sun et al., 2025; Li et al., 2026). Memory-centric designs (Shi et al., 2025; Sridhar et al., 2025) further store recent observations or key frames to support short-term reasoning. Despite this progress, most existing VLAs remain fundamentally view-dependent, lacking mechanisms to retain and reuse spatial information beyond the current camera frustum. Consequently, they struggle with manipulation when task-relevant objects are temporarily unseen. This work addresses this limitation by introducing a spatial-grounded memory that enables persistent scene understanding and reliable out-of-vision manipulation, expanding the effective operational scope of VLA systems.

# 3. Method

As shown in Figure 2, SOMA addresses the challenge of out-of-vision manipulation by equipping VLA with a spatial memory that supports persistent scene understanding beyond the current camera view. Prior to manipulation, if the target object referenced in the instruction cannot be localized in the current camera observation by the perception module, the robot performs a dedicated scene observation phase using a movable head camera to initialize *Spatial Memory Construction*. Given a multi-view scanning se-quence, object-level semantics and their relative spatial relationships are extracted by combining efficient visual perception and geometric modeling. These representations are aggregated into an overview scene memory $\mathcal{M}_0$, which encodes a unified spatial–semantic representation of all observed objects. During manipulation, the model receives the current observation $o_c^t$, the user instruction, robot states, and a noised action sequence, where $c \in \{l, r, h\}$ denotes the left arm, right arm, and head cameras. New observations from the head view $o_h^t$ are incorporated to update $\mathcal{M}_0$ into $\hat{\mathcal{M}}_t$ through *Dynamic Memory Refinement*, which performs similarity-aware fusion to preserve global consistency while accommodating newly observed evidence. Subsequently, *Contextual Memory Retrieval* aligns instruction-grounded vision–language embeddings with $\hat{\mathcal{M}}_t$, selectively activating task-relevant spatial representations to augment the current perceptual features. The resulting memory-enhanced vision–language tokens, together with robot states and noised action embeddings, are processed by DiT blocks and an action decoder to predict the next action chunk. By maintaining a globally consistent spatial memory, SOMA enables robust reasoning and manipulation even when task-relevant objects lie outside the current field of view.

## 3.1. Spatial Memory Construction

*Spatial Memory Construction* builds a unified spatial–semantic map of the environment by integrating object-level semantics and 3D geometry from head-camera exploration, serving as a global memory foundation for subsequent perception and manipulation. Given a scanning video sequence $V = \{f_i\}_{i=1}^{N_v}$, we uniformly sample one frame every $N$ frames to obtain a subset $\tilde{V}$, which is used to construct the overview scene memory $\mathcal{M}_0$. Each sampled frame $f_i \in \tilde{V}$ is processed by a unified perception pipeline consisting of: (1) a geometry prior network (VGGT (Wang et al., 2025b)) for camera pose and coarse scene geometry estimation, (2) a semantic perception module (YOLO (Cheng et al., 2024)) for object localization and categorization, and (3) a semantic encoder (DINOv3 (Siméoni et al., 2025)) for high-dimensional visual feature extraction. Given the per-frame feature map $\mathbf{F}^{(i)} \in \mathbb{R}^{H \times W \times C}$, each detected 2D bounding box $b_j^{(i)}$ is used to crop its corresponding region, followed by spatial average pooling to obtain an instance-level appearance embedding $\mathbf{f}_j^{(i)} \in \mathbb{R}^C$. Each 2D detection is further lifted into the global 3D scene coordinate system using VGGT-predicted geometric priors, yielding a 3D bounding box $\mathbf{b}_j^{(i)} \in \mathbb{R}^{8 \times 3}$. VGGT predicts the camera pose of each frame in a consistent scene-level coordinate system, which we treat as a shared global reference frame. All lifted 3D bounding boxes are expressed in this global frame using the estimated camera poses, enabling direct aggregation and comparison across views. Since the head-camera scan is short-horizon and performed in a static environment, pose

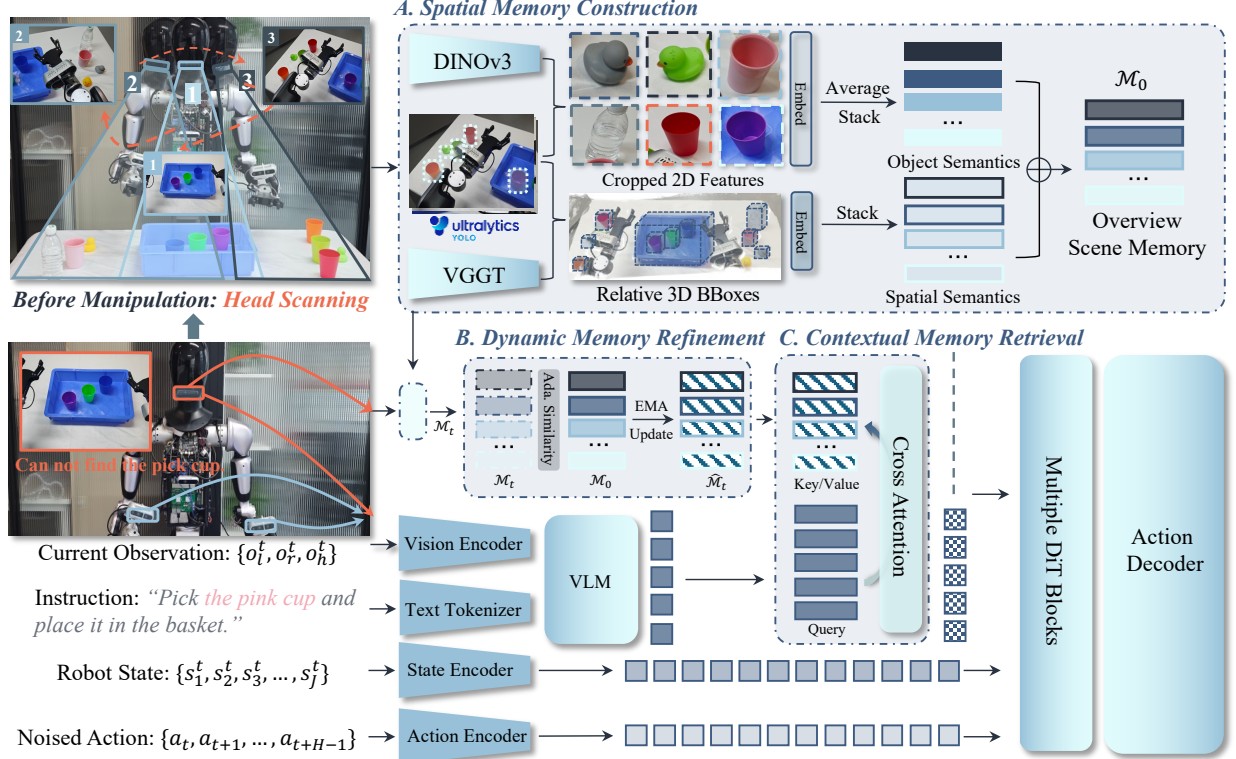

*Figure 2.* Illustration of the proposed SOMA framework. SOMA enhances OOV manipulation via spatial memory. (A) **Spatial Memory Construction:** Before manipulation, if the specified objects are not in the current observation, the robot actively scans the scene to construct a unified spatial–semantic memory $\mathcal{M}_0$ by integrating object semantics (YOLO (Cheng et al., 2024), DINOv3 (Siméoni et al., 2025)) with geometric cues (VGGT (Wang et al., 2025b)). (B) **Dynamic Memory Refinement:** During interaction, newly perceived information $\mathcal{M}_t$ is adaptively fused into the initial overview scene memory $\mathcal{M}_0$ through similarity-weighted updates, yielding a refined current memory $\hat{\mathcal{M}}_t$. (C) **Contextual Memory Retrieval:** Language-guided queries selectively attend to relevant memory regions, providing contextual cues for a DiT-based Action Decoder to generate executable manipulation actions.

drift is negligible in practice and does not affect instance association or memory construction.

Across all sampled frames $\tilde{V}$, detected instances are associated and fused across views to form a global set of view-invariant object instances. To resolve same-class ambiguities and multi-view duplicates, we perform class-wise instance association based on a joint appearance–geometry similarity, combining cosine similarity between DINOv3 embeddings and 3D spatial consistency between lifted bounding boxes. Instances with similarity above a threshold are merged, and their appearance features and 3D geometries are aggregated by averaging. This process yields global collections $\mathcal{F} = \{\mathbf{f}_k\}_{k=1}^{N_I}$, $\mathcal{B}_{3D} = \{\mathbf{b}_k\}_{k=1}^{N_I}$, and $\mathcal{C}_{3D} = \{c_k\}_{k=1}^{N_I}$, where $N_I$ denotes the number of fused object instances. To improve robustness under imperfect detections or missing observations, learned placeholder embeddings and pseudo bounding boxes are injected when no objects are detected in a sampled frame. Each global instance is ultimately represented by a unified triplet $(\mathbf{f}_k, c_k, \mathbf{b}_k)$, forming the fine-grained basis for constructing the overview scene memory.

To construct a unified scene memory that captures both semantic identity and geometric configuration, we integrate object-level appearance embeddings with their corresponding 3D spatial encodings. Each 3D bounding box $\mathbf{b}_k$ is embedded into a spatial positional vector $\mathbf{p}_k = \Phi_{pos}(\mathbf{b}_k)$, where $\Phi_{pos}(\cdot)$ denotes the learnable 3D positional embedding module. The object features $\mathbf{f}_k$ are then projected into a shared embedding space through a learnable mapping $\Phi_{mem}(\cdot)$, and combined with their corresponding spatial descriptors to form the joint spatial–semantic embedding: $\mathbf{m}_k^0 = \Phi_{mem}(\mathbf{f}_k) + \mathbf{p}_k$. The resulting $\mathbf{m}_k^0 \in \mathbb{R}^C$ serves as the final memory token, jointly encoding appearance semantics and 3D spatial context. Collectively, the set of all such embeddings forms the overview scene memory: $\mathcal{M}_0 = \{\mathbf{m}_k^0\}_{k=1}^{N_I}$, which provides a compact, geometry-aware, and semantically discriminative representation of the entire scene.

### 3.2. Dynamic Memory Refinement

During manipulation, the environment evolves as objects move, become occluded, or newly appear in the scene.

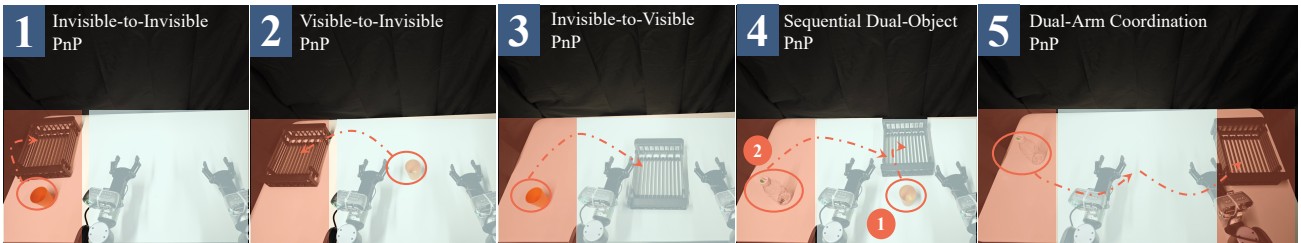

*Figure 3.* Illustration of our real world benchmark settings. We design five challenging *out-of-vision* pick-and-place (PnP) tasks to evaluate the robot's OOV manipulation capabilities. Tasks (1–3) require the robot to use its left arm to pick up the object located outside the current field of view and place it into the basket. Task (4) involves sequential pick-and-place of the object followed by the other object into the same basket. Task (5) requires dual-arm coordination to pick and place the object into the basket.

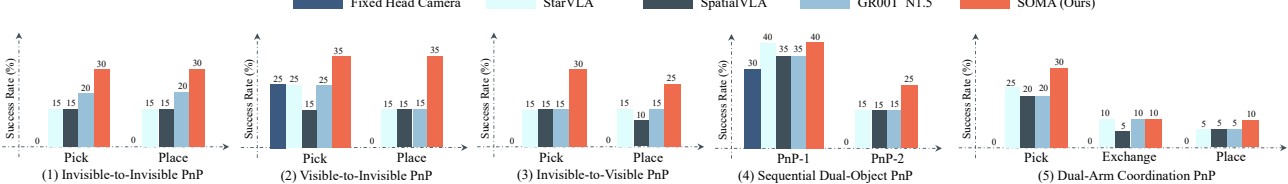

*Figure 4.* Performance comparison across five real world out-of-vision tasks. "Fixed Head Camera" denotes we train SOMA under fixed head camera setting. StarVLA (Ye et al., 2026; Community, 2026), SpatialVLA (Qu et al., 2025),GR00T N1.5 (Bjorck et al., 2025) and our proposed SOMA is trained under active head camera for fair. We adopt 20 episodes and average SR (Success Rate) to evaluate the models by multi-stages.

| Behavioral Analysis on Real-World Out-of-Vision Tasks | | | | | |
|---|---|---|---|---|---|
| Model | Task 1 | Task 2 | Task 3 | Task 4 | Task 5 |
| *First-Fixation Time (s)* ↓ | | | | | |
| GR00T-N1.5 | 7.6 | 21.0 | 14.8 | 10.9 | 11.5 |
| SOMA | $4.2_{\downarrow 45\%}$ | $12.7_{\downarrow 40\%}$ | $8.2_{\downarrow 45\%}$ | $4.9_{\downarrow 55\%}$ | $4.7_{\downarrow 59\%}$ |
| *Head Search Path Length (deg)* ↓ | | | | | |
| GR00T-N1.5 | 50.5 | 51.0 | 83.8 | 109.2 | 164.0 |
| SOMA | $27.8_{\downarrow 45\%}$ | $28.1_{\downarrow 45\%}$ | $50.3_{\downarrow 40\%}$ | $54.6_{\downarrow 50\%}$ | $70.4_{\downarrow 57\%}$ |
| *Viewpoint Correction Count* ↓ | | | | | |
| GR00T-N1.5 | 1.6 | 1.9 | 1.4 | 3.4 | 5.3 |
| SOMA | $0.9_{\downarrow 44\%}$ | $1.1_{\downarrow 42\%}$ | $0.8_{\downarrow 43\%}$ | $1.7_{\downarrow 50\%}$ | $2.3_{\downarrow 57\%}$ |
| *Grasp Attempt Count* ↓ | | | | | |
| GR00T-N1.5 | 1.8 | 2.0 | 1.7 | 2.4 | 3.7 |
| SOMA | $1.0_{\downarrow 44\%}$ | $1.2_{\downarrow 40\%}$ | $1.0_{\downarrow 41\%}$ | $1.2_{\downarrow 50\%}$ | $1.6_{\downarrow 57\%}$ |
| *Time-to-Grasp (s)* ↓ | | | | | |
| GR00T-N1.5 | 58.0 | 30.0 | 50.0 | 65.5 | 36.5 |
| SOMA | $32.3_{\downarrow 44\%}$ | $16.8_{\downarrow 44\%}$ | $29.7_{\downarrow 41\%}$ | $30.4_{\downarrow 54\%}$ | $14.6_{\downarrow 60\%}$ |

*Table 1.* Behavioral comparison between GR00T-N1.5 and SOMA on five real-world OOV manipulation tasks.

To maintain an up-to-date and globally coherent scene representation, *Dynamic Memory Refinement* is proposed to iteratively update the scene memory initialized by the overview memory $\mathcal{M}_0$ using the latest observation from the robot's head-view camera. The current observation $o_h^t$ is processed by the same embedding pipeline described in Section 3.1 at time $t$, producing the current memory tokens $\mathcal{M}_t = \{\mathbf{m}_j^t\}_{j=1}^{N_t}$, where each $\mathbf{m}_j^t = \Phi_{\text{mem}}(\mathbf{f}_j^t) + \bar{\mathbf{p}}_{c_j^t}$ jointly encodes semantic and spatial cues extracted from the current frame, and $\bar{\mathbf{p}}_{c_j^t}$ denotes the positional embedding derived from the estimated 3D location of the observed instance.

For each observed instance $\mathbf{m}_j^t$ with class label $c_j^t$, we perform instance-level association against the memory entries

of the same class in the previous memory state $\mathcal{M}_{t-1}$. If a matching memory instance $\mathbf{m}_k^{t-1}$ is found using the same class-wise appearance–geometry association criterion as in *Spatial Memory Construction,* its representation is refined through an *adaptive similarity-aware fusion mechanism.* Each observation is associated to at most one memory instance, and each memory entry is updated by at most one observation per time step $t$. Given the previous memory vector $\mathbf{m}_k^{t-1}$ and the new observation $\mathbf{m}_j^t$, we compute the semantic similarity $s_{kj}^t$ and a dynamic fusion score $g_{kj}^t$:

$$s_{kj}^t = \sigma\big(\Phi_{\text{sim}}([\mathbf{m}_k^{t-1} - \mathbf{m}_j^t])\big), g_{kj}^t = \sigma\big(\Phi_{\text{fuse}}([\mathbf{m}_k^{t-1}, \mathbf{m}_j^t])\big), \tag{1}$$

where $\sigma(\cdot)$ denotes the sigmoid activation. The two scores jointly determine an adaptive update coefficient $\alpha_{kj}^t = g_{kj}^t \cdot s_{kj}^t$. The matched memory token is then updated via a temporal exponential moving average:

$$\mathbf{m}_k^t = \alpha_{kj}^t \mathbf{m}_j^t + (1 - \alpha_{kj}^t) \mathbf{m}_k^{t-1}. \tag{2}$$

This design adaptively balances new visual evidence with historical memory, allowing the representation to remain stable under minor viewpoint changes while rapidly adapting to meaningful scene updates. Observed instances without matched counterparts are appended as new memory entries, while unmatched past memory instances are retained to handle temporary occlusions and partial observability. The updated memory is denoted as $\hat{\mathcal{M}}_t = \{\hat{\mathbf{m}}_k^t\}_{k=1}^{N_t^M}$, where $N_t^M$ is the number of memory entries. This instance-aware temporal refinement process ensures memory consistency across time, faithfully reflecting the evolving scene state.

### 3.3. Contextual Memory Retrieval

We inject global spatial–semantic knowledge into the multi-modal representation for instruction-grounded reasoning. Given the vision–language token embeddings from the VLM, $\mathbf{X}_{\mathrm{vl}} = \{\mathbf{x}_i \in \mathbb{R}^C\}_{i=1}^{N_q}$, where $N_q$ is the token number and each $\mathbf{x}_i$ represents a visual–linguistic feature. We treat $\mathbf{X}_{\mathrm{vl}}$ as the query set, and the scene memory, $\hat{\mathcal{M}}_t$ as the key–value memory bank containing the structured spatial context. Each memory token $\mathbf{m}_k^t$ is first projected into the same latent space as the VLM features via an alignment function: $\tilde{\mathbf{m}}_k^t = \Phi_{\mathrm{align}}(\hat{\mathbf{m}}_k^t)$. The retrieval process is realized through a cross-attention module, where $\mathbf{X}_{\mathrm{vl}}$ serves as the query $\mathbf{Q}$ and $\tilde{\mathcal{M}}_t$ as the key–value pair $\mathbf{K}, \mathbf{V}$:
$\mathbf{X}_{\mathrm{boost}} = \{\mathbf{x}_i'\}_{i=1}^{N_q} = \mathrm{softmax}\left(\frac{\mathbf{Q}\mathbf{K}^\top}{\sqrt{C}}\right)\mathbf{V}$, which $\mathbf{x}_i'$ denotes the enhanced token representations enriched with spatially grounded memory cues. Through this scenario knowledge interaction, each token $\mathbf{x}_i$ selectively attends to relevant regions in $\tilde{\mathcal{M}}_t$, retrieving geometry-aware information that bridges current perception with global scene context.

Finally, the retrieved memory-enhanced features $\mathbf{X}_{\mathrm{boost}}$ are injected into the DiT blocks as a spatial–semantic prior. The original vision–language tokens, robot state, and noised action embeddings are directly fed into the DiT, where $\mathbf{X}_{\mathrm{boost}}$ serves as global context that modulates token interactions. Through these DiT blocks, the model fuses current observations with the structured scene memory, producing refined state and action tokens. These refined tokens are then passed to the embodiment-specific action decoder to generate the final action chunk $\{\hat{a}_t, \hat{a}_{t+1}, \dots, \hat{a}_{t+H-1}\}$, where $H$ denotes the action chunk length.

## 4. Experiments

### 4.1. Benchmarks

We conduct extensive real-world experiments to validate the effectiveness of SOMA in OOV manipulation. We further evaluate its memory-centric design on RoboCasa Tabletop GR1 (Bjorck et al., 2025) and SimplerEnv (Li et al., 2024).

**Real World OOV Tasks.** As shown in Figure 3 and 5, we construct a real-world benchmark of five out-of-vision pick-and-place (PnP) tasks with increasing difficulty to evaluate SOMA under partial observability. All tasks require manipulating objects outside the robot's initial field of view, directly probing the model's ability to operate beyond instantaneous visual input. Tasks (1)–(3) evaluate single-object manipulation under different visibility transitions, testing spatial recall and re-identification. Task (4) introduces sequential dual-object manipulation, requiring spatial memory to be preserved and reused across stages. Task (5) further involves dual-arm coordination, where success depends on maintaining a globally consistent spatial representation. These tasks

| OOV Task | Scan+GR00T | No-Scan SOMA | Scan-only SOMA | Full SOMA |
|---|---|---|---|---|
| Task 1 | 19.0 | 20.0 | 25.0 | **30.0** |
| Task 2 | 22.0 | 24.0 | 29.0 | **35.0** |
| Task 3 | 16.0 | 17.5 | 22.5 | **27.5** |
| Task 4 | 25.0 | 26.0 | 30.0 | **32.5** |
| Task 5 | 10.5 | 11.7 | 14.2 | **16.7** |
| SR (%) | 18.5 | 19.8 | 24.1 | **28.3** |

*Table 2.* Ablation study on scan-based exploration and spatial memory for real-world OOV manipulation. *Scan+GR00T* performs head scanning and uses the detected target frame as a fixed visual input for a reactive policy, without maintaining a persistent spatial memory. *No-Scan SOMA* initializes the memory from the first observed frame without multi-view scanning. *Scan-only SOMA* constructs the memory from multi-view scans but disables memory refinement during manipulation.

enable a behavior-level evaluation of spatial memory, revealing its impact on target localization, viewpoint efficiency, and manipulation under partial observability.

**Static Simulation Tasks.** We evaluate the memory design of SOMA on RoboCasa Tabletop GR1 (Nasiriany et al., 2024; Bjorck et al., 2025) and SimplerEnv. RoboCasa Tabletop GR1 provides a large-scale tabletop manipulation suite with diverse objects, layouts, and multi-stage tasks requiring joint grounding of vision and language. SimplerEnv offers a standardized real-to-sim benchmark for evaluating policy success rates across simulated environments reflecting real-world robotic systems (Zitkovich et al., 2023).

### 4.2. Implementation

**Robot Setting.** Real-world experiments are conducted on a self-designed humanoid platform equipped with two 7-DoF Realman ZM73 arms and adaptive grippers. The robot features a 2-DoF motorized head with pan–tilt control, enabling viewpoint adjustment during manipulation. For multi-view perception, Intel RealSense D435 RGB-D cameras are mounted on the head and both wrists, providing synchronized visual and depth observations. We also deploy an NVIDIA Jetson AGX Orin (64 GB) on robot.

**Training Details.** To accelerate training, *Spatial Memory Construction* is performed offline during data preprocessing. For real-world experiments, we collect 400 expert demonstrations per task using a VR teleoperation system. Each trajectory consists of a scanning phase, used to construct the overview scene memory, and a manipulation phase, which is used for training all comparison models. For simulation experiments, the overview memory is constructed from the first frame of each trajectory. "Full" denotes training on the complete RoboCasa Tabletop dataset. We adopt GR00T N1.5 (Bjorck et al., 2025) as the real-world baseline. During training, all components are optimized except the VLM language decoder, using multi-task learning with a batch size of 60 for 30,000 steps on 32 NVIDIA H200 GPUs.

**Inference Details.** All inference are executed on a server equipped with an NVIDIA RTX 4090 GPU. In real-world

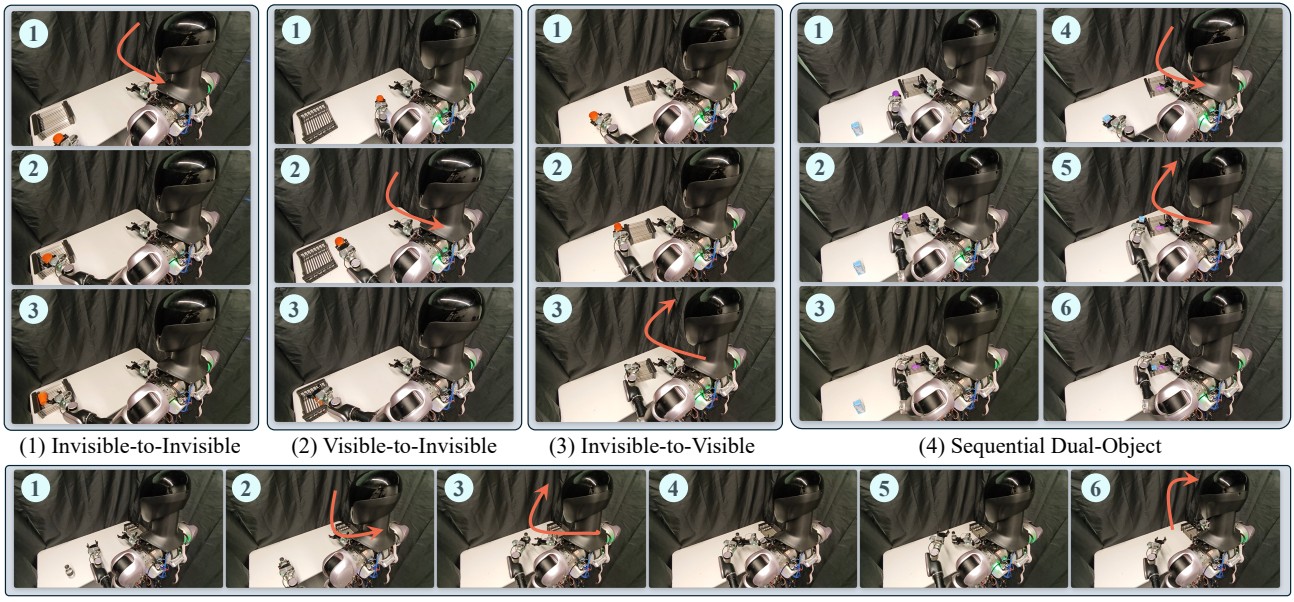

(1) Invisible-to-Invisible     (2) Visible-to-Invisible     (3) Invisible-to-Visible     (4) Sequential Dual-Object

(5) Dual-Arm Coordination

*Figure 5.* Illustration of task execution examples for our five challenging out-of-vision tasks in real world using our proposed SOMA.

experiments, the *Spatial Memory Construction* phase is triggered by a perception-based criterion: a lightweight object detector checks whether the target specified in the instruction is visible in the current view. If the target cannot be localized, SOMA initiates an active head-scanning procedure along a predefined trajectory to construct the spatial memory. In simulation, the overview memory is constructed from the initial robot observation.

### 4.3. Real World Results

In Figure 4, SOMA achieves the highest success rates across all five real-world out-of-vision (OOV) manipulation tasks. The fixed-head variant fails once either the target or the goal leaves the field of view, confirming the brittleness of view-bound policies under partial observability. Although prior VLAs with active head control (e.g., StarVLA (Ye et al., 2026; Community, 2026), SpatialVLA (Qu et al., 2025), and GR00T-N1.5 (Bjorck et al., 2025)) can partially recover visibility, their performance degrades in multi-stage and dual-arm tasks, where spatial information must be preserved and reused across stages. In contrast, SOMA maintains consistently higher success rates across both Pick and Place stages, with the performance gap widening as task complexity increases. These results indicate that persistent spatial memory, rather than reactive viewpoint adjustment, is critical for reliable manipulation beyond the current visual frustum.

Moreover, we also investigate the different behaviour mode between out SOMA and GR00T as shown in Table 1. It demonstrates that SOMA's advantages go beyond improved

success rates and manifest as qualitatively different execution behavior. Across all five tasks, SOMA consistently shortens the time required to visually acquire the target, reduces head search motion, and stabilizes viewpoint alignment, achieving 40–60% improvements over GR00T-N1.5 depending on task difficulty. In particular, SOMA exhibits near one-shot grasping behavior, with significantly fewer grasp attempts and corrective head motions, indicating more decisive and spatially grounded execution. The behavioral gap becomes increasingly pronounced as task difficulty increases. On the most challenging dual-arm Task 5, SOMA substantially reduces head search effort, viewpoint corrections, and time-to-grasp, highlighting its ability to maintain a consistent global spatial representation across multi-arm coordination and staged manipulation. These results demonstrate that SOMA's spatial memory enables stable cross-stage spatial grounding, leading to both higher efficiency and greater reliability in out-of-vision manipulation.

Finally, Table 2 disentangles the roles of scan-based exploration and explicit spatial memory in real-world out-of-vision manipulation. *Scan+GR00T*, which performs head scanning without maintaining a persistent spatial memory, yields the lowest performance, indicating that scan-based exploration alone is insufficient for reliable OOV manipulation. *No-Scan SOMA* slightly outperforms Scan+GR00T despite using only a single-view initialization, highlighting the benefit of an explicit memory structure even without multi-view coverage. *Scan-only SOMA* further improves success rates by leveraging multi-view scanning to construct a more complete initial memory, but still falls short of the full model. In contrast, *Full SOMA* consistently achieves the

| Task Category | Diffusion Policy (Chi et al., 2025) | | | | StarVLA* (Community, 2026) | | | | GR00T N1.5 (Bjorck et al., 2025) | | | | SOMA (Ours) | | | |
|---|---|---|---|---|---|---|---|---|---|---|---|---|---|---|---|---|
| | 30 | 100 | 300 | Full | 30 | 100 | 300 | Full | 30 | 100 | 300 | Full | 30 | 100 | 300 | Full |
| Container Interaction | 35.1 | 49.5 | 54.2 | - | 1.4 | 1.1 | 1.8 | 3.0 | 35.3 | 33.7 | 35.7 | 40.3 | 52.3 | 52.0 | 53.3 | 55.6 |
| Cooking Preparation | 19.2 | 31.4 | 35.3 | - | 10.5 | 14.2 | 25.8 | 27.6 | 40.8 | 45.2 | 42.8 | 49.2 | 43.6 | 50.8 | 48.4 | 46.4 |
| Tabletop Serving | 13.2 | 18.4 | 28.4 | - | 15.0 | 18.5 | 24.0 | 25.0 | 41.0 | 47.5 | 48.5 | 39.5 | 44.5 | 46.5 | 54.0 | 47.5 |
| Dish Transfer | 16.9 | 22.3 | 39.0 | - | 11.3 | 15.0 | 21.5 | 22.5 | 44.0 | 58.5 | 51.0 | 50.0 | 58.0 | 58.0 | 55.0 | 49.5 |
| Tray Organization | 15.7 | 32.6 | 39.2 | - | 25.3 | 28.2 | 27.9 | 28.8 | 36.0 | 40.0 | 43.6 | 50.8 | 43.2 | 38.0 | 48.8 | 47.6 |
| **Average** | 20.0 | 30.8 | 39.2 | - | 12.7 | 15.4 | 20.2 | 21.4 | 39.4 | 44.9 | 44.3 | 45.9 | **48.3** | **49.1** | **52.0** | **49.3** |

*Table 3.* Performance comparison via SR (%) across task categories on the Robocasa Tabletop GR1 benchmarks with varying numbers of demonstrations per task. Each group of four columns corresponds to one method (30 / 100 / 300 / Full demos per task). Bold numbers indicate the best result for each setting. * denotes that we utilize the Qwen-GR00T setting for development.

| (a) Visual Matching | | | | |
|---|---|---|---|---|
| Model | Pick Coke Can | Move Near | Open/Close Drawer | Average |
| Moto (Chen et al., 2025e) | 74.0 | 60.4 | 43.1 | 59.2 |
| RoboVLM (Liu et al., 2025) | 77.3 | 61.7 | 43.5 | 60.6 |
| TraceVLA (Zheng et al., 2024) | 28.0 | 53.7 | 57.0 | 42.0 |
| $\pi_0$ (Intelligence et al., 2025) | 72.7 | 65.3 | 38.3 | 58.8 |
| $\pi_0$+FAST (Driess et al., 2025) | 75.3 | 67.5 | 42.9 | 61.9 |
| OpenVLA-OFT (Kim et al., 2025) | 72.3 | 69.6 | 47.2 | 63.0 |
| GR00T-N1.5 (Bjorck et al., 2025) | 47.0 | 70.0 | 18.1 | 45.0 |
| SOMA (Ours) | **85.0** | 73.0 | 31.5 | **63.2** |
| (b) Variant Aggregation | | | | |
| Model | Pick Coke Can | Move Near | Open/Close Drawer | Average |
| OpenVLA (Kim et al., 2024) | 54.5 | 47.7 | 17.7 | 39.8 |
| RoboVLM (Liu et al., 2025) | **75.6** | 60.0 | 10.6 | 51.3 |
| TraceVLA (Zheng et al., 2024) | 60.0 | 56.4 | **31.0** | 45.0 |
| OpenVLA-OFT (Kim et al., 2025) | 65.3 | 59.0 | 12.2 | 45.5 |
| GR00T-N1.5 (Bjorck et al., 2025) | 46.7 | 62.9 | 17.5 | 42.4 |
| SOMA (Ours) | 55.5 | **76.6** | 25.4 | **52.5** |

*Table 4.* SimplerEnv evaluation across different SOTAs on Google Robot tasks. We report results of all models pretrained with OXE (O'Neill et al., 2024) dataset and then fine-tuned with Fractal dataset (Brohan et al., 2022).

best performance by continuously refining and interacting with the spatial memory during manipulation. These results demonstrate that while scanning can provide useful coverage for bootstrapping, the dominant performance gains arise from persistent spatial memory and its ongoing refinement, rather than scanning itself.

### 4.4. Simulation Results

**GR1 Results.** Table 3 presents SR across five task categories under varying numbers of demonstrations. Overall, SOMA achieves the highest average performance of 52.0% with 300 demos and maintains competitive results across all data regimes, surpassing strong baselines such as Diffusion Policy (Chi et al., 2025), GR00T (Bjorck et al., 2025), and StarVLA (Community, 2026). The advantage is particularly evident in *Dish Transfer* and *Tabletop Serving*, where SOMA consistently outperforms diffusion-based policies by over 15%, demonstrating the effectiveness of its designed spatial memory in handling multi-stage, spatially complex manipulations. Notably, even with only 30–100 demonstrations per task, SOMA maintains high success rates, revealing strong sample efficiency and robust generalization to unseen spatial configurations. These consistent gains across all categories confirm that grounding perception and action generation in a structured spatial memory yields more coherent, globally aware manipulation in complex, multi-object environments.

| Ablation Setting | w/o Geo. | w/o Obj. | w/o Update | Full (Ours) |
|---|---|---|---|---|
| Geometric cues | | ✓ | ✓ | ✓ |
| Object Semantics | ✓ | | ✓ | ✓ |
| Dynamic Update | ✓ | ✓ | | ✓ |
| **Overall Success (%)** | 45.1 | 43.7 | 41.5 | **49.3** |

*Table 5.* Ablation study on different components of the proposed memory design. "Geo." and "Obj." denote Geometric cues and object semantics, respectively.

**SimplerEnv Results.** Table 4 reports the performance comparison across multiple SOTA VLAs. In the *Visual Matching* setting (Table 4(a)), our SOMA achieves the best SR of 63.2%, surpassing all listed SOTAs. Compared to strong baselines like RoboVLM (Liu et al., 2025) (60.6%), SOMA improves performance particularly on challenging partial-observation tasks like *Pick Coke Can* (+7.7%), highlighting the benefit of its designed memory mechanism. In the *Variant Aggregation* setting (Table 4(b)), SOMA also achieves the highest overall average of 52.5%, demonstrating strong robustness and spatial consistency across task variants.

### 4.5. Ablation Study

We utilize GR1 benchmark for evaluation and leverage *full training version* as the baseline model. All experiments evaluated over 50 episodes for accuracy. *More detailed results are provided in the Appendix.*

**Overall Memory Construction.** As shown in Table 5, we conduct the ablation study on different components of the overview scene memory. Removing positional cues (*w/o Geo.*) or object semantics (*w/o Obj.*) leads to clear performance drops from 49.3% to 45.1% and 43.7%, respectively, confirming that both spatial geometry and semantic identity are indispensable for constructing a coherent global memory. Further excluding the dynamic update mechanism (*w/o Update*) causes the largest degradation to 41.5%, highlighting the importance of temporal refinement in maintaining memory consistency across observations.

## 5. Conclusion

We propose SOMA, a spatial memory framework for Vision-Language-Action models that addresses the fundamental limitation of view-bound perception in out-of-vision manip-

ulation. By constructing, refining, and retrieving a persistent spatial–semantic memory from multi-view observations, SOMA enables robots to retain and reuse spatial evidence beyond instantaneous visual input. This memory-centric design supports robust reasoning and manipulation when task-relevant objects are temporarily outside the field of view, without relying on brittle view-dependent behaviors. Extensive real-world experiments demonstrate the practical effectiveness of SOMA in challenging out-of-vision manipulation scenarios, while additional evaluations in simulation show that the proposed memory mechanism remains beneficial under conventional fully observable settings.

## Acknowledgements

This work was supported in part by the National Key R&D Program of China (Grant No.2023YFF0725001), in part by the National Natural Science Foundation of China (Grant No.92370204),in part by the guangdong Basic and Applied Basic Research Foundation (Grant No.2023B1515120057), in part by the Key-Area Special Project of Guangdong Provincial Ordinary Universities (2024ZDZX1007).

## Impact Statement

This work addresses the challenge of manipulation under limited operational field of view by introducing a spatial memory framework that enables robots to reason about and act on objects beyond the current camera view. By integrating multi-view scene exploration, persistent spatial memory, and contextual interaction, the proposed approach improves robustness in long-horizon manipulation tasks where relevant objects cannot be directly observed at execution time. This capability is particularly relevant for real-world robotic applications such as household assistance and industrial manipulation, where sensing is often constrained by embodiment and task geometry. The framework is modular and builds on standard perception components, allowing future improvements in perception and safety-aware control to further enhance reliability.

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

# A. Data Collection

## A.1. VR Teleoperation System

Our robot is as shown in Figure. 6. To collect high-quality human demonstration data, we develop a VR-driven full-body teleoperation system using the Meta Oculus Quest 3 as the unified interface for controlling both robot arms and the active head as shown in Figure 7. The Quest 3 provides precise 6-DoF tracking and responsive controller input, enabling natural and coordinated bimanual operation. For arm manipulation, the operator wears a full-scale bimanual exoskeleton, inspired by GELLO (Wu et al., 2024), which establishes a joint-to-joint kinematic mapping from the operator to the robot's dual 7-DoF humanoid arms. The Quest 3 tracks the operator's global pose and controller actions, which are fused with exoskeleton data in the control workstation to generate smooth and human-like trajectories. For head control, the Quest 3 is likewise used as the main interaction device. Instead of streaming head pose or performing inside-VR rendering (Xiong et al., 2025; Zeng et al., 2025), we map the robot head's pan–tilt actions to designated Quest 3 controller buttons, allowing the operator to adjust the robot's viewpoint in a stable, discrete, and low-latency manner. Crucially, to ensure minimal latency, high stability, and accurate visual alignment, we do not stream any robot camera views or rendered UI into the VR headset (Zeng et al., 2025; Xiong et al., 2025). All visual feedback—including wrist-mounted and head-mounted RGB-D camera feeds—is displayed exclusively on a monitor connected to an external teleoperation workstation. During demonstration collection, both the robot and the Quest 3 connect directly to this workstation, which acts as a unified hub for sensor fusion, control synchronization, and real-time visualization. This design avoids the additional encoding, wireless streaming, and rendering delays that VR-in-headset display would introduce, resulting in a significantly more responsive teleoperation loop. This VR-assisted but monitor-viewed teleoperation architecture provides a stable, low-latency, and highly controllable setup for collecting precise, human-aligned manipulation demonstrations.

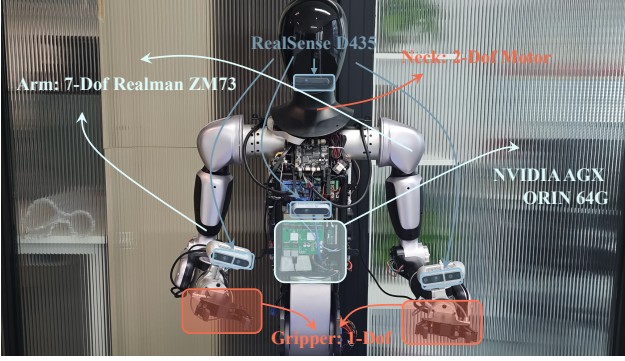

*Figure 6.* Illustration of our self-designed robot construction.

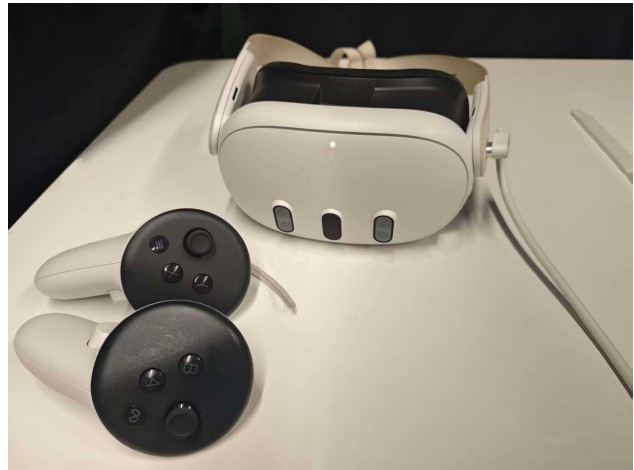

*Figure 7.* Illustration of the designed VR teleoperation System using Meta Oculus Quest 3.

## A.2. Real World Data Quality Review

To ensure the reliability of teleoperated demonstrations collected in real-world settings, we conduct a frame-by-frame, trajectory-level data quality review. This manual inspection is designed to filter out sequences affected by latency, sensor artifacts, operator mistakes, or camera capture inconsistencies. We define a set of deterministic rules to evaluate each trajectory:

*1) Excessive Actuation Latency.* Both gripper and joints commands are monitored for temporal alignment. Any sequence exhibiting ≥300 ms delay between user input and recorded robot actuation is marked invalid, as such latency typically arises from communication bottlenecks or onboard processing delays and leads to distorted motion supervision.

*2) Unintentional Gripper Oscillation.* Gripper trajectories containing rapid, continuous open–close toggling are flagged as operator mis-triggers and discarded, as these patterns do not reflect meaningful manipulation intent.

*3) Gripper Value Spikes.* The gripper sensor normally outputs values within the 0–1000 range. We label as corrupted any trajectory containing systematic spiking (*e.g.*, 1001–1010) caused by low-level hardware protocol noise or overflow. Such discontinuities compromise the smoothness required for imitation learning.

*4) Camera View Contamination.* Demonstrations are rejected if the robot cameras capture large human-body intrusions, missing robot arms, or severe motion blur from aggressive operator movement. These issues hinder visual–kinematic alignment and degrade learning.

*5) Invalid Reach Strategy.* For humanoid-arm grasping, we enforce a consistent approach strategy. Specifically, trajectories employing top-down, table-type grasping are filtered

out, as such motions contradict the intended horizontal, in-hand-level grasping typical for humanoid robots and lead to inconsistent action patterns.

*6) Video–Joint Stream Misalignment (Frame Dropping).* Raw RGB-D recordings are capped at 25 FPS. Any sequence with $\geq 2.5$ dropped frames is removed to avoid misalignment between visual frames and joint trajectories, which is critical for paired multimodal supervision.

Across all criteria, the review ensures that only temporally aligned, semantically valid, and visually consistent demonstrations enter the training set, forming a stable foundation for robust real-world robot learning.

### A.3. Real-World Data Statistics

The real-world evaluation benchmark comprises five challenging *out-of-vision* Pick-and-Place (PnP) tasks, designed to systematically evaluate manipulation under partial observability. The five tasks include *Invisible-to-Invisible PnP*, *Visible-to-Invisible PnP*, *Invisible-to-Visible PnP*, *Sequential Dual-Object PnP*, and *Dual-Arm Coordination PnP*, each emphasizing a different form of spatial memory usage.

*1) Invisible-to-Invisible PnP* requires the robot to grasp a target object that is outside the current field of view and place it at a target location that is also outside the field of view, testing the ability to operate entirely beyond direct visual observation.

*2) Visible-to-Invisible PnP* involves grasping a target object that is initially visible and placing it at a target location outside the current view, evaluating whether spatial information about unseen goal locations can be recalled and utilized.

*3) Invisible-to-Visible PnP* requires the robot to grasp an object that is initially outside the field of view and place it at a visible target location, testing spatial recall of unseen objects followed by precise execution.

*4) Sequential Dual-Object PnP* extends the setting to multi-stage manipulation: the robot first performs pick-and-place for a visible object, and subsequently grasps and places a second object located outside the field of view, stressing memory persistence across sequential task stages.

*5) Dual-Arm Coordination PnP* represents the most challenging scenario, where both the target object and target placement location are outside the field of view, and successful execution requires coordinated dual-arm handover followed by accurate placement based on a globally consistent spatial representation.

Each task involves three distinct objects and was collected with 400 expert demonstrations. The raw data is downsampled by a factor of two, resulting in 711,524 frames at an average frame rate of 12.74 Hz, corresponding to approxi-

mately 15.52 hours of expert demonstrations. It provides a structured real-world benchmark for evaluating spatial memory and manipulation behavior when both objects and goals may lie outside the robot's immediate field of view.

## B. Methods

### B.1. Data Preprocess

As we mentioned in Section 4.2, we accelerate the whole training process by abstract the pipeline of semantics processing by utilizing the series of external powerful perception models (Wang et al., 2025b; Cheng et al., 2024; Siméoni et al., 2025) as shown in Algorithm 1. In detailed, To efficiently construct the overview scene memory used by *SOMA*, we adopt an offline preprocessing pipeline that extracts spatial–semantic evidence from both the scanning phase and the manipulation phase of each trajectory. For each trajectory $\tau$, the process begins by generating the overview memory from the head-camera scanning video. Given a scanning sequence $V^{(\tau)}$, we uniformly sample frames every $(N/3)$ steps to obtain a reduced subset $\tilde{V}^{(\tau)} = \{f_i \mid i \bmod (N/3) = 0\}$ that balances spatial coverage with computational efficiency. Each sampled frame is then processed through a unified perception stack—the geometry prior $\Phi_{\text{geo}}$, the object detector $\mathcal{Y}$, and the semantic encoder $\Phi_{\text{sem}}$, which extract instance-level embeddings $\mathbf{f}_j^{(i)}$, categorical labels $c_j^{(i)}$, and their corresponding 3D bounding boxes $\mathbf{b}_j^{(i)}$. These multi-view instances are aggregated across the entire scanning sequence and fused into the initial overview instance record $\mathcal{R}_{\text{sparse}}^{(\tau)}[0]$, which provides a compact yet globally consistent summary of the scene.

After establishing the overview memory, we process the manipulation segment of each trajectory. Instead of extracting features at every timestep—which would incur excessive computational overhead—we again apply uniform subsampling, selecting only frames whose manipulation step index satisfies $t \bmod N = 0$. For each such sampled frame $f_t$, we run the same perception pipeline to obtain the triplet $(\mathbf{f}^{(t)}, c^{(t)}, \mathbf{b}^{(t)})$, storing it in the sparse record $\mathcal{R}_{\text{sparse}}^{(\tau)}[t]$. This strategy captures the temporal evolution of key task-relevant objects while significantly reducing preprocessing cost, and empirically preserves all information needed for effective memory refinement during policy execution.

Finally, to ensure that every timestep in the trajectory is paired with a valid memory observation during training, we perform a back-mapping procedure that expands the sparse record into a dense per-step memory sequence. For each timestep $t$ in the full trajectory, we identify the nearest sampled index $t' = \lfloor t/N \rfloor \cdot N$ and assign its stored evidence to the current step, i.e., $\mathcal{R}_t^{(\tau)} \leftarrow \mathcal{R}_{\text{sparse}}^{(\tau)}[t']$. This

**Algorithm 1** MEMORYPREPROCESS: Offline Extraction of Spatial–Semantic Evidence

---

**Input:** Dataset $\mathcal{D}$, sampling interval $N$, batch size $B$, geometry prior $\Phi_{\text{geo}}$, detector $\mathcal{Y}$, semantic encoder $\Phi_{\text{sem}}$

**Output:** Per-step memory evidence $\{\mathcal{R}_t^{(\tau)}\}$ for all trajectories

**for** each trajectory $\tau$ in $\mathcal{D}$ **do**

  Initialize sparse record $\mathcal{R}_{\text{sparse}}^{(\tau)} \leftarrow \emptyset$

  **(1) Overview Memory from Scanning Video ($t{=}0$)**

  Sample overview frames $\tilde{V}^{(\tau)} = \{f_i \mid i \bmod (N/3) = 0\}$

  **for** each batch $\{f_i\}$ of size $B$ in $\tilde{V}^{(\tau)}$ **do**

    Extract object features $(\mathbf{F}, \mathcal{C}_{\text{obj}}) \leftarrow \Phi_{\text{sem}}, \mathcal{Y}(f_i)$

    Lift boxes to 3D $\mathcal{B}_{3D} \leftarrow \Phi_{\text{geo}}(f_i)$

    Accumulate instance triplets $(\mathbf{f}_j^{(i)}, c_j^{(i)}, \mathbf{b}_j^{(i)})$

  **end for**

  Fuse all instances

  $\mathcal{R}_{\text{sparse}}^{(\tau)}[0] \leftarrow \text{FUSE}(\{\mathbf{f}_j^{(i)}\})$

  **(2) Per-step Frame Sampling (Manipulation)**

  Sample manipulation steps $\tilde{S}^{(\tau)} = \{t \mid t \bmod N = 0\}$

  **for** each $t \in \tilde{S}^{(\tau)}$ with $t > 0$ **do**

    Process frame $f_t$

    $(\mathbf{f}^{(t)}, c^{(t)}, \mathbf{b}^{(t)}) \leftarrow \Phi_{\text{sem}}, \mathcal{Y}, \Phi_{\text{geo}}(f_t)$

    Store sparse evidence

    $\mathcal{R}_{\text{sparse}}^{(\tau)}[t] \leftarrow (\mathbf{f}^{(t)}, c^{(t)}, \mathbf{b}^{(t)})$

  **end for**

  **(3) Back-mapping to All Steps**

  **for** each step $t$ in trajectory $\tau$ **do**

    Find closest sampled step $t' = \lfloor t/N \rfloor \cdot N$

    $\mathcal{R}_t^{(\tau)} \leftarrow \mathcal{R}_{\text{sparse}}^{(\tau)}[t']$

  **end for**

**end for**

**Return:** $\{\mathcal{R}_t^{(\tau)}\}$ for all trajectories

---

back-mapping guarantees temporal consistency and avoids missing-memory cases, while keeping the computational load tied only to the sampled frames. The resulting dense instance record $\{\mathcal{R}_t^{(\tau)}\}$ serves as the input for *Spatial Memory Construction* during training, enabling **SOMA** to learn manipulation policies grounded on structured, geometry-aware, and temporally coherent environmental context.

### B.2. Framework Details

We employs a set of purpose-built neural modules that support geometric encoding, semantic abstraction, temporal fusion, and memory–VLM interaction, all of which are essential for enabling SOMA's spatial memory–driven OOV manipulation pipeline, as summarized in Table 6.

**3D BBox Embedding MLP.** This module transforms each

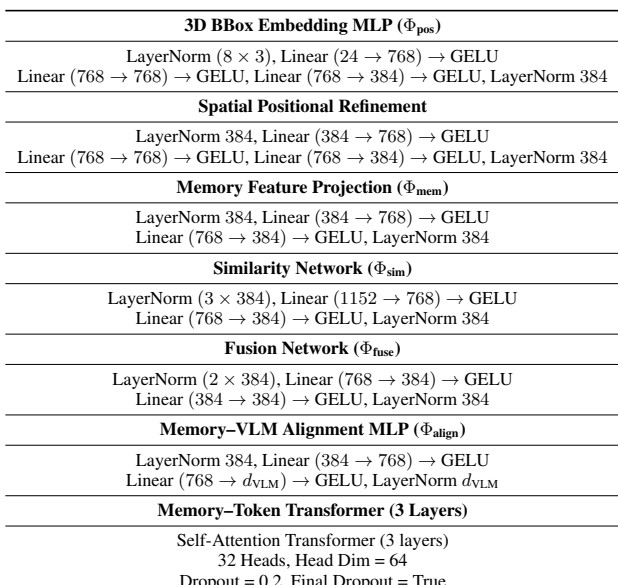

| **3D BBox Embedding MLP ($\Phi_{\text{pos}}$)** |
| :---: |
| LayerNorm ($8 \times 3$), Linear ($24 \rightarrow 768$) $\rightarrow$ GELU |
| Linear ($768 \rightarrow 768$) $\rightarrow$ GELU, Linear ($768 \rightarrow 384$) $\rightarrow$ GELU, LayerNorm $384$ |
| **Spatial Positional Refinement** |
| LayerNorm $384$, Linear ($384 \rightarrow 768$) $\rightarrow$ GELU |
| Linear ($768 \rightarrow 768$) $\rightarrow$ GELU, Linear ($768 \rightarrow 384$) $\rightarrow$ GELU, LayerNorm $384$ |
| **Memory Feature Projection ($\Phi_{\text{mem}}$)** |
| LayerNorm $384$, Linear ($384 \rightarrow 768$) $\rightarrow$ GELU |
| Linear ($768 \rightarrow 384$) $\rightarrow$ GELU, LayerNorm $384$ |
| **Similarity Network ($\Phi_{\text{sim}}$)** |
| LayerNorm ($3 \times 384$), Linear ($1152 \rightarrow 768$) $\rightarrow$ GELU |
| Linear ($768 \rightarrow 384$) $\rightarrow$ GELU, LayerNorm $384$ |
| **Fusion Network ($\Phi_{\text{fuse}}$)** |
| LayerNorm ($2 \times 384$), Linear ($768 \rightarrow 384$) $\rightarrow$ GELU |
| Linear ($384 \rightarrow 384$) $\rightarrow$ GELU, LayerNorm $384$ |
| **Memory–VLM Alignment MLP ($\Phi_{\text{align}}$)** |
| LayerNorm $384$, Linear ($384 \rightarrow 768$) $\rightarrow$ GELU |
| Linear ($768 \rightarrow d_{\text{VLM}}$) $\rightarrow$ GELU, LayerNorm $d_{\text{VLM}}$ |
| **Memory–Token Transformer (3 Layers)** |
| Self-Attention Transformer (3 layers) |
| 32 Heads, Head Dim = 64 |
| Dropout = 0.2, Final Dropout = True |

*Table 6.* Architectures of the modules used for spatial–semantic embedding, memory refinement, and contextual retrieval.

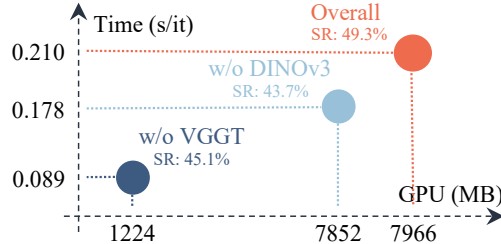

*Figure 8.* Analysis about time and gpu cost per demo of different component choice in *the memory preprocess stage*.

raw 3D bounding box $\mathbf{b}_k \in \mathbb{R}^{8 \times 3}$ into a stable spatial descriptor $\mathbf{p}_k = \Phi_{\text{pos}}(\mathbf{b}_k)$. It is used in *Spatial Memory Construction* to encode viewpoint-invariant geometry, ensuring that all instances are anchored in a consistent spatial reference frame.

**Spatial Positional Refinement.** A second-stage MLP refines $\mathbf{p}_k$ to suppress geometric noise from multi-view lifting. This step increases the reliability of positional embeddings before they are fused into the memory tokens.

**Memory Feature Projection.** The projection module $\Phi_{\text{mem}}$ maps visual embeddings $\mathbf{f}_k$ into the memory space and combines them with $\mathbf{p}_k$ to form the unified memory tokens $\mathbf{m}_k^0$. This module establishes a consistent representation format for *Spatial Memory Construction* and ensures that semantics and geometry are jointly encoded.

**Similarity and Fusion MLP.** During *Dynamic Memory Refinement*, two compact MLPs compute a semantic similarity score $s_{kj}^t$ and a fusion confidence score $g_{kj}^t$ between historical memory and new observations. Their product defines an adaptive update weight $\alpha_{kj}^t$, enabling selective memory

| Category | Details |
|---|---|
| **Container Interaction** | **Tasks:** Cup→Drawer, Potato/Milk→Microwave, Bottle/Wine/Can→Cabinet. **Summary:** Multi-stage pick–place–close interactions with drawers, cabinets, or microwaves, evaluating sequential manipulation and spatial planning. |
| **Cooking Preparation** | **Tasks:** Cuttingboard→Basket/Pan/Pot/Tieredbasket. **Summary:** Transfers ingredients from a cutting board into cookware, testing grasp stability and pre-cooking preparation skills. |
| **Tabletop Serving** | **Tasks:** Placemat→Bowl/Plate/Tieredshelf. **Summary:** Serving-like actions placing items from a placemat to tableware; emphasizes precise placement and tabletop spatial reasoning. |
| **Dish Transfer** | **Tasks:** Plate→Bowl/Box/Pan/Plate. **Summary:** Transfers items between dishware during dining or cleanup, evaluating fine-grained control and container-to-container reasoning. |
| **Tray Organization** | **Tasks:** Tray→Plate/Pot/Tieredbasket/Tieredshelf. **Summary:** Organizes items from a tray into various structured storages, testing multi-target placement and organization planning. |

*Table 7.* Summary of task categories in the Robocasa Tabletop GR1 benchmark.

| | Task Name | # Variations |
|---|---|---|
| *Bridge* | Spoon On Tower | 1 |
| | Carrot On Plate | 1 |
| | Stack Cube | 1 |
| | Eggplant In Basket | 1 |
| *Fractal* | Pick Coke Can | VM: 12 / VA: 33 |
| | Move Near | VM: 4 / VA: 10 |
| | Open/Close Drawer | VM: 216 / VA: 42 |

*Table 8.* SimplerEnv task details. "VM" denotes visual-matching and "VA" denotes variant-aggregation. Bridge tasks use a single configuration; Fractal tasks report VM and VA separately.

refinement that is stable under minor viewpoint change yet responsive to true scene updates.

**Memory–VLM Alignment MLP.** This module projects memory tokens into the VLM backbone's feature space. It provides the key–value representation for *Contextual Memory Retrieval*, ensuring that the memory can be directly queried by instruction-conditioned visual–linguistic tokens.

**Memory–Token Transformer.** A lightweight attention transformer performs cross-modal interaction between the aligned memory and VLM tokens. This component injects global spatial–semantic structure into the VLM representation, producing the memory-boosted feature $\mathbf{X}_{\text{boost}}$ that conditions the downstream DiT action predictor.

## C. Realworld Evaluation

As shown in Figure 9, we add more example with different object manipulation on our self-designed out-of-vision tasks including (1) Sequential Dual-Object and (2) Dual-Arm Coordination.

## D. Simulation Experiments

### D.1. Benchmarks

**Robocasa Tabletop GR1.** This dataset provides a digital analogue to real-world humanoid demonstrations, enabling systematic evaluation prior to deployment on physical robots (Bjorck et al., 2025; Nasiriany et al., 2024). It features tabletop rearrangement tasks executed by the GR-1 humanoid robot equipped with Fourier dexterous hands, emphasizing dexterous hand–arm coordination under diverse tabletop settings. The GR-1 benchmark contains a substantially broader variety of objects, receptacles, and spatial configurations. As shown in Table 7, the dataset consists of 24 manipulation tasks, which can be organized into the following categories based on the underlying interaction structure:

*1) Container Interaction Tasks* (e.g., placing items into cabinets, drawers, or microwaves and subsequently closing them), testing multi-stage pick–place–close behaviors with articulated receptacles.

*2) Cooking Preparation Tasks* (e.g., transferring objects from a cutting board into pans, pots, or baskets), simulating preparatory cooking steps that require stable grasping and reliable object placement.

*3) Tabletop Serving Tasks* (e.g., moving items from a placemat to bowls or plates), capturing serving-style behaviors that depend on precise spatial reasoning.

*4) Dish Transfer Tasks* (e.g., moving objects between dishware such as plate→bowl), evaluating fine-grained coordination during mid-meal or cleanup scenarios.

*5) Tray Organization Tasks* (e.g., transporting items from a tray to multi-level shelves or containers), assessing multi-target placement and organized spatial planning.

| Task Category | Diffusion Policy (Chi et al., 2025) | | | | GR00T N1.5 (Bjorck et al., 2025) | | | | SOMA (Ours) | | | |
|---|---|---|---|---|---|---|---|---|---|---|---|---|
| | 30 | 100 | 300 | Full | 30 | 100 | 300 | Full | 30 | 100 | 300 | Full |
| CupToDrawerClose | 24.5 | 32.4 | 36.3 | – | 34.0 | 44.0 | 44.0 | 44.0 | 60.0 | 46.0 | 46.0 | 60.0 |
| PotatoToMicrowaveClose | 17.7 | 30.4 | 41.2 | – | 24.0 | 26.0 | 32.0 | 28.0 | 36.0 | 44.0 | 54.0 | 34.0 |
| MilkToMicrowaveClose | 37.3 | 41.2 | 51.0 | – | 58.0 | 46.0 | 54.0 | 44.0 | 54.0 | 44.0 | 55.0 | 64.0 |
| BottleToCabinetClose | 40.2 | 62.8 | 60.8 | – | 44.0 | 22.0 | 32.0 | 40.0 | 62.0 | 72.0 | 80.0 | 64.0 |
| WineToCabinetClose | 43.1 | 55.9 | 60.8 | – | 28.0 | 34.0 | 34.0 | 44.0 | 52.0 | 52.0 | 56.0 | 64.0 |
| CanToDrawerClose | 48.0 | 74.5 | 75.5 | – | 24.0 | 33.0 | 36.0 | 42.0 | 50.0 | 54.0 | 58.0 | 54.0 |
| **Container Interaction** | 35.1 | 49.5 | 54.3 | – | 35.3 | 33.7 | 35.7 | 40.3 | 52.3 | 52.0 | 53.3 | 55.7 |
| CuttingboardToBasket | 19.6 | 42.2 | 29.4 | – | 38.0 | 42.0 | 50.0 | 42.0 | 32.0 | 56.0 | 56.0 | 38.0 |
| CuttingboardToCardboardbox | 11.8 | 15.7 | 22.6 | – | 36.0 | 40.0 | 50.0 | 36.0 | 48.0 | 40.0 | 32.0 | 44.0 |
| CuttingboardToPan | 28.4 | 48.0 | 57.8 | – | 62.0 | 68.0 | 46.0 | 74.0 | 56.0 | 50.0 | 52.0 | 62.0 |
| CuttingboardToPot | 22.6 | 37.3 | 48.0 | – | 48.0 | 40.0 | 46.0 | 74.0 | 46.0 | 46.0 | 52.0 | 42.0 |
| CuttingboardToTieredbasket | 13.7 | 13.7 | 18.6 | – | 20.0 | 36.0 | 28.0 | 52.0 | 36.0 | 62.0 | 50.0 | 48.0 |
| **Cooking Preparation** | 19.2 | 31.4 | 35.3 | – | 40.8 | 45.2 | 42.8 | 49.2 | 43.6 | 50.8 | 48.4 | 46.4 |
| PlacematToBasket | 15.7 | 25.5 | 41.2 | – | 38.0 | 46.0 | 42.0 | 34.0 | 36.0 | 48.0 | 44.0 | 42.0 |
| PlacematToBowl | 14.7 | 18.6 | 23.5 | – | 52.0 | 54.0 | 62.0 | 28.0 | 64.0 | 50.0 | 58.0 | 54.0 |
| PlacematToPlate | 15.7 | 23.5 | 37.3 | – | 54.0 | 52.0 | 52.0 | 56.0 | 54.0 | 54.0 | 56.0 | 72.0 |
| PlacematToTieredshelf | 6.9 | 5.9 | 11.8 | – | 20.0 | 28.0 | 38.0 | 56.0 | 24.0 | 34.0 | 58.0 | 22.0 |
| **Tabletop Serving** | 13.2 | 18.4 | 28.4 | – | 41.0 | 47.5 | 48.5 | 39.5 | 44.5 | 46.5 | 54.0 | 47.5 |
| PlateToBowl | 15.7 | 18.6 | 31.4 | – | 48.0 | 50.0 | 44.0 | 26.0 | 62.0 | 56.0 | 52.0 | 48.0 |
| PlateToCardboardbox | 12.8 | 13.7 | 27.5 | – | 32.0 | 52.0 | 52.0 | 54.0 | 46.0 | 52.0 | 54.0 | 34.0 |
| PlateToPan | 13.7 | 17.7 | 35.3 | – | 48.0 | 60.0 | 38.0 | 46.0 | 54.0 | 58.0 | 46.0 | 68.0 |
| FromPlateToPlate | 25.5 | 39.2 | 61.8 | – | 50.0 | 72.0 | 38.0 | 74.0 | 70.0 | 66.0 | 68.0 | 48.0 |
| **Dish Transfer** | 16.9 | 22.3 | 39.0 | – | 44.0 | 58.5 | 51.0 | 50.0 | 58.0 | 58.0 | 55.0 | 49.5 |
| TrayToCardboardbox | 21.6 | 37.3 | 40.2 | – | 32.0 | 40.0 | 44.0 | 66.0 | 46.0 | 34.0 | 56.0 | 60.0 |
| TrayToPlate | 26.5 | 41.2 | 49.0 | – | 54.0 | 58.0 | 46.0 | 44.0 | 52.0 | 52.0 | 58.0 | 68.0 |
| TrayToPot | 21.6 | 48.0 | 52.9 | – | 42.0 | 44.0 | 46.0 | 52.0 | 52.0 | 32.0 | 56.0 | 40.0 |
| TrayToTieredbasket | 12.8 | 34.3 | 39.2 | – | 30.0 | 34.0 | 56.0 | 46.0 | 50.0 | 50.0 | 52.0 | 48.0 |
| TrayToTieredshelf | 2.0 | 6.9 | 15.7 | – | 22.0 | 24.0 | 26.0 | 30.0 | 16.0 | 22.0 | 22.0 | 32.0 |
| **Tray Organization** | 15.7 | 32.6 | 39.2 | – | 36.0 | 40.0 | 43.6 | 50.8 | 43.2 | 38.0 | 48.8 | 47.6 |
| **Average** | 20.1 | 30.8 | 39.2 | – | 39.4 | 45.0 | 44.3 | 46.0 | 48.3 | 49.1 | 51.9 | 49.3 |

*Table 9.* Performance comparison via success rate (%) across task categories on the Robocasa Tabletop GR-1 benchmark with varying numbers of demonstrations per task. Diffusion Policy is only evaluated under 30/100/300 settings.

Most tasks involve distractor objects and unseen combinations of source–target receptacles, requiring the policy to interpret task language and adapt to novel rearrangement structures. The tasks closely follow the rearrangement pattern introduced in pre-training—moving objects from a source region to a target receptacle—yet deployment settings and object pairings are deliberately held out to test generalization. The observation space consists of a single egocentric RGB view obtained from the robot's head-mounted camera. The state space includes the 3D joint positions and orientations of both arms and dexterous hands, as well as waist and neck control. End-effector–based actions are also provided as an alternative control interface for whole-body IK.

**SimplerEnv.** As shown in Table 8, the SimplerEnv benchmark comprises two complementary suites: Bridge and Fractal. The Bridge suite contains four tabletop manipulation tasks executed on a WidowX robot—*Spoon on Towel, Carrot on Plate, Stack Cube, and Eggplant in Basket.* Each task is associated with a single language instruction template and centers on core object-placement and stacking primitives, offering a clean, controlled evaluation setup. The Fractal suite builds on the RT-1 dataset collected with the Google robot and defines three additional tasks: *Pick Coke Can, Move Near, Open/Close Drawer.* Each task is evaluated under two protocols. Visual Matching (VM) approximates real-world deployment by varying object configurations and URDF assets to ensure consistency between simulation and physical execution. Variant Aggregation (VA) introduces extensive visual perturbations—spanning background changes, texture shifts, lighting variations, distractors, and camera viewpoint alterations—to rigorously stress-test robustness and generalization. *In this paper, we utilize the Fractal suite to assess our memory design.*

### D.2. Behavioral Metric Definitions

To better understand how spatial memory influences manipulation behavior under partial observability, we report a set of behavioral metrics in Table 1. All metrics are computed per episode and then averaged over 20 evaluation episodes for each task.

*1) First-Fixation Time.* First-Fixation Time measures the temporal latency between the moment the target object first enters the head camera's field of view and the initiation of the robot's grasping motion. Specifically, it is defined as the

| Task Category | Update Strategy | | | Retrieval Module | | | | Memory Representation | | | Full |
|---|---|---|---|---|---|---|---|---|---|---|---|
| | SimEMA | Only $s_{kj}^t$ | Only $g_{kj}^t$ | Concat | Light | Heavy | Middle | Obj. | Geo. | NoUpd | |
| CupToDrawerClose | 25.0 | 25.0 | 25.0 | 50.0 | 42.0 | 43.0 | 60.0 | 35.0 | 45.0 | 45.0 | 60.0 |
| PotatoToMicrowaveClose | 25.0 | 50.0 | 20.0 | 20.0 | 25.0 | 25.0 | 34.0 | 30.0 | 20.0 | 25.0 | 34.0 |
| MilkToMicrowaveClose | 60.0 | 45.0 | 50.0 | 35.0 | 30.0 | 30.0 | 64.0 | 60.0 | 40.0 | 45.0 | 64.0 |
| BottleToCabinetClose | 20.0 | 45.0 | 20.0 | 40.0 | 50.0 | 40.0 | 64.0 | 30.0 | 40.0 | 30.0 | 64.0 |
| WineToCabinetClose | 45.0 | 25.0 | 35.0 | 35.0 | 35.0 | 35.0 | 54.0 | 45.0 | 25.0 | 20.0 | 54.0 |
| CanToDrawerClose | 15.0 | 25.0 | 25.0 | 25.0 | 40.0 | 30.0 | 58.0 | 25.0 | 30.0 | 25.0 | 58.0 |
| **Container Interaction** | 31.7 | 35.8 | 29.2 | 34.2 | 37.0 | 33.8 | 55.7 | 37.5 | 33.3 | 31.7 | 55.7 |
| CuttingboardToBasket | 55.0 | 50.0 | 50.0 | 25.0 | 35.0 | 45.0 | 38.0 | 45.0 | 35.0 | 40.0 | 38.0 |
| CuttingboardToCardboardbox | 60.0 | 40.0 | 40.0 | 45.0 | 50.0 | 60.0 | 44.0 | 35.0 | 55.0 | 25.0 | 44.0 |
| CuttingboardToPan | 65.0 | 65.0 | 50.0 | 55.0 | 60.0 | 45.0 | 60.0 | 70.0 | 75.0 | 60.0 | 60.0 |
| CuttingboardToPot | 55.0 | 50.0 | 50.0 | 65.0 | 40.0 | 60.0 | 42.0 | 55.0 | 60.0 | 65.0 | 42.0 |
| CuttingboardToTieredbasket | 25.0 | 40.0 | 45.0 | 30.0 | 30.0 | 30.0 | 48.0 | 15.0 | 25.0 | 40.0 | 48.0 |
| **Cooking Preparation** | 52.0 | 49.0 | 47.0 | 44.0 | 43.0 | 48.0 | 46.4 | 44.0 | 50.0 | 46.0 | 46.4 |
| PlacematToBasket | 50.0 | 15.0 | 50.0 | 45.0 | 35.0 | 40.0 | 42.0 | 40.0 | 45.0 | 45.0 | 42.0 |
| PlacematToBowl | 35.0 | 55.0 | 60.0 | 70.0 | 65.0 | 70.0 | 54.0 | 45.0 | 50.0 | 45.0 | 54.0 |
| PlacematToPlate | 35.0 | 55.0 | 80.0 | 60.0 | 50.0 | 60.0 | 72.0 | 65.0 | 50.0 | 40.0 | 72.0 |
| PlacematToTieredshelf | 10.0 | 10.0 | 25.0 | 40.0 | 10.0 | 25.0 | 22.0 | 20.0 | 35.0 | 25.0 | 22.0 |
| **Tabletop Serving** | 32.5 | 33.8 | 53.8 | 53.8 | 40.0 | 48.8 | 47.5 | 42.5 | 45.0 | 38.8 | 47.5 |
| PlateToBowl | 35.0 | 55.0 | 50.0 | 55.0 | 40.0 | 55.0 | 48.0 | 60.0 | 45.0 | 45.0 | 48.0 |
| PlateToCardboardbox | 35.0 | 25.0 | 55.0 | 40.0 | 55.0 | 55.0 | 34.0 | 45.0 | 50.0 | 35.0 | 34.0 |
| PlateToPan | 30.0 | 50.0 | 50.0 | 50.0 | 40.0 | 55.0 | 58.0 | 50.0 | 35.0 | 50.0 | 58.0 |
| FromPlateToPlate | 65.0 | 50.0 | 60.0 | 60.0 | 60.0 | 65.0 | 58.0 | 70.0 | 45.0 | 70.0 | 58.0 |
| **Dish Transfer** | 41.2 | 45.0 | 53.8 | 51.2 | 48.8 | 57.5 | 49.5 | 56.2 | 43.7 | 50.0 | 49.5 |
| TrayToCardboardbox | 40.0 | 35.0 | 55.0 | 50.0 | 50.0 | 40.0 | 60.0 | 55.0 | 35.0 | 40.0 | 60.0 |
| TrayToPlate | 55.0 | 60.0 | 45.0 | 65.0 | 50.0 | 50.0 | 68.0 | 55.0 | 70.0 | 50.0 | 68.0 |
| TrayToPot | 55.0 | 35.0 | 60.0 | 55.0 | 45.0 | 50.0 | 40.0 | 45.0 | 50.0 | 40.0 | 40.0 |
| TrayToTieredbasket | 45.0 | 45.0 | 40.0 | 35.0 | 60.0 | 20.0 | 48.0 | 40.0 | 45.0 | 40.0 | 48.0 |
| TrayToTieredshelf | 35.0 | 20.0 | 5.0 | 25.0 | 25.0 | 15.0 | 22.0 | 40.0 | 20.0 | 35.0 | 22.0 |
| **Tray Organization** | 47.5 | 40.0 | 37.5 | 45.0 | 45.0 | 33.7 | 47.6 | 45.0 | 46.2 | 41.2 | 47.6 |
| **Average** | **41.0** | **40.7** | **44.2** | **45.6** | **42.8** | **44.4** | **49.3** | **45.1** | **43.7** | **41.5** | **49.3** |

*Table 10.* Detailed Ablation studies on Robocasa Tabletop GR-1 benchmark. We compare different Update Strategies, Retrieval Modules, and Memory Representations. Reported values are success rates (%). "SimEMA" denotes the normal EMA update (Chen et al., 2020), $s_{kj}^t$ denotes semantic similarity, $g_{kj}^t$ denotes dynamic fusion scores, "Concat" denotes naive fusion of perception tokens and memory features "Light/Middle/Heavy" use 2/3/4 cross-attention layers. "Geo." and "Obj." denote Geometric cues and object semantics, respectively. "NoUpd" denotes the memory representation without updating with latest observations.

| Task Category | Frame Sampling Interval | | | |
|---|---|---|---|---|
| | 10 | 20 | 30 | 40 |
| Container Interaction | 41.7 | **55.7** | 38.7 | 35.7 |
| Cuttingboard-Origin | 45.6 | **46.4** | 45.2 | 45.2 |
| Placemat-Origin | 45.5 | **47.5** | 44.5 | 44.0 |
| Plate-Origin | 50.5 | 49.5 | 50.0 | **52.0** |
| Tray-Origin | 41.0 | **47.6** | 46.0 | 45.2 |
| **Average** | 44.9 | **49.3** | 44.9 | 44.4 |

*Table 11.* Ablation Study of different processing interval choices in *Spatial Memory Construction*. The interval controls the frame sampling frequency from scanning sequences.

duration (in seconds) from the first frame in which the target becomes visually observable to the timestamp at which the arm begins the grasp execution. This value is determined through offline human annotation of recorded trajectories to ensure high-fidelity temporal accuracy independent of the robot's control frequency. Lower values indicate a faster transition from visual acquisition to physical manipulation,

reflecting more decisive and well-grounded behavior.

*2) Head Search Path Length.* Head Search Path Length measures the total amount of head motion required to bring the target object into view. Let $t \in \{0, 1, \ldots, T-1\}$ index environment steps in an episode, and let $(\psi_t, \phi_t)$ denote the head pan (yaw) and tilt (pitch) angles at step $t$, measured in degrees. Let $t_{\text{vis}}$ denote the step at which the target object first becomes visible in the head camera view, as determined by offline annotation. The Head Search Path Length is defined as the cumulative angular displacement of the head prior to target visibility:

$$L_{\text{head}} = \sum_{t=1}^{t_{\text{vis}}} \left( |\psi_t - \psi_{t-1}| + |\phi_t - \phi_{t-1}| \right). \quad (3)$$

Smaller values indicate more efficient viewpoint search with less unnecessary head motion.

*3) Viewpoint Correction Count.* Viewpoint Correction Count counts the number of discrete head reorientation events performed after the target object has already been observed

at least once. Each correction corresponds to a change in head motion direction intended to re-center or re-acquire the target. This metric captures the stability of target localization, where fewer corrections imply more consistent spatial grounding.

*4) Grasp Attempt Count.* Grasp Attempt Count measures the number of grasp executions issued by the controller until a successful grasp is achieved. Each grasp attempt is defined as a completed gripper closing action. Lower values indicate more reliable target localization and manipulation, with values close to one corresponding to near one-shot grasping behavior.

*5) Time-to-Grasp.* Time-to-Grasp measures the total number of environment steps from episode start until a successful grasp is completed. This metric aggregates both perception and manipulation efficiency, reflecting the overall speed of task execution under partial observability.

### D.3. Quantitative Results

As shown in Table 9, SOMA achieves SOTA performance on the RoboCasa Tabletop GR-1 benchmark across all data regimes. Importantly, this benchmark operates under static observability, where all task-relevant objects are visible, allowing us to isolate the effect of the spatial memory design independent of viewpoint acquisition. SOMA attains the highest overall average success rate, reaching 49.3% under the full-data setting, outperforming both GR00T-N1.5 (46.0%) and Diffusion Policy (39.2% under 300-shot). Beyond absolute performance, SOMA exhibits markedly improved sample efficiency. With only 30 demonstrations per task, SOMA already achieves an average success rate of 48.3%, surpassing GR00T-N1.5 trained on the full dataset. This advantage is consistent across task categories, with particularly strong gains in *Container Interaction* (55.7% vs. 40.3% for GR00T-N1.5 under full data), which involves multi-step manipulation and structured object–receptacle relationships. These results indicate that SOMA's spatial memory—through explicit scene representation and instruction-conditioned retrieval—provides a more structured inductive bias for manipulation. Even when all objects are directly observable, maintaining and querying a global spatial–semantic memory leads to more stable learning and better generalization than purely reactive policies.

### D.4. Qualitative Results

The qualitative results of *GR1 benchmark* (Bjorck et al., 2025; Nasiriany et al., 2024) and *SimplerEnv Fractal* (Li et al., 2024; Zitkovich et al., 2023) are as shown in Figure (10,11).

### D.5. Detailed Ablation Study

We further expand our ablation analysis from the coarse summary in Table (5) to a fine-grained breakdown in Table 10, complemented by the efficiency study in Fig. 8 and the sampling-interval analysis in Table 11. Together, these results substantiate the effectiveness of SOMA by isolating the gains from its key designs—*Dynamic Memory Refinement*, *Contextual Memory Retrieval*, and the spatial–semantic memory representation—while also clarifying the associated computational trade-offs and the impact of memory construction frequency.

**Dynamic Memory Refinement.** Comparing the *Update Strategy* variants, the *Full* method (Average: 49.3%) significantly outperforms the baselines (SimEMA: 41.0%, Only $s_{kj}^t$: 40.7%, Only $g_{kj}^t$: 44.2%). This demonstrates that the complete adaptive similarity-aware fusion mechanism—which leverages the product of semantic similarity ($s_{kj}^t$) and learned dynamic fusion score ($g_{kj}^t$), $\alpha_{kj}^t = g_{kj}^t \cdot s_{kj}^t$—is essential for maintaining global memory consistency and temporal coherence. Specifically, relying solely on semantic similarity ($s_{kj}^t$) or dynamic confidence ($g_{kj}^t$) results in a performance drop of approximately $8.6\% \sim 8.3\%$ in average success rate, confirming that the *joint, adaptive update coefficient* is the most effective strategy for stabilizing the memory under continuous visual changes.

**Contextual Memory Retrieval.** The performance of the *Retrieval Module* shows a strong correlation with model capacity. The final *Full* (Average: 49.3%) utilizes the *Middle* module which employs a sufficient number of cross-attention layers. The simpler *Light* module (42.8%) and the naive *Concat* fusion (45.6%) perform substantially worse. Although *Heavy* achieves a competitive 44.4%, the *Middle* config achieves the optimal balance between performance and architectural complexity, suggesting that three cross-attention layers are adequate for effectively integrating the queried scene memory ($\hat{\mathcal{M}}_t$) into the VLM's perception tokens to achieve instruction-grounded spatial reasoning.

**Memory Representation.** Analysis of the *Memory Representation* confirms the necessity of integrating both geometric and semantic priors. The full representation (Full: 49.3%)—which couples appearance features ($\mathbf{f}_k$) with 3D spatial encodings ($\mathbf{p}_k$)—significantly outperforms variants that omit critical information: *Geo.* (43.7%): Omitting object semantics (relying only on geometry/VLM features) leads to a 5.6% performance drop, highlighting that discriminative semantics (from YOLO/DINOv3) are crucial for robust object identification. *Obj.* (45.1%): Omitting 3D geometric cues (relying only on object features) results in a 4.2% drop, emphasizing that *geometric grounding* (from VGGT) is indispensable for precise spatial reasoning and action generation in 3D space. Finally, *NoUpd* (41.5%): Using the initial memory $\mathcal{M}_0$ without the *Dynamic Mem-*

| Method | Inference Latency (s / chunk) |
|---|---|
| StarVLA (Community, 2026) | ∼1.26 |
| SpatialVLA (Qu et al., 2025) | ∼1.45 |
| GR00T-N1.5 (Bjorck et al., 2025) | ∼1.30 |
| SOMA (Ours) | ∼1.58 |

*Table 12.* Comparison of online inference latency across diverse VLA methods on real-world experiments.

| Geometry Source | OOV SR (%) ↑ | Latency (s) ↓ |
|---|---|---|
| RGB-D + OpenCV fusion | 22.6 | 0.18 |
| VGGT geometry + pose | 27.3 | 0.21 |

*Table 13.* Ablation on geometry source and alignment strategy. We compare RGB-D based point cloud fusion with VGGT-predicted geometry in terms of OOV task performance and computational overhead.

| Failure Mode | Count | Prop. | Description |
|---|---|---|---|
| Inaccurate Grasp Pose Under Dynamic Viewpoint | 12/25 | 48.0% | Shifting head-camera views during visual search degrade grasp estimation, causing misaligned approach or gripper slippage (most pronounced in Tasks 1, 3, 5 with extended head movement). |
| Placement Timing and Precision Errors | 8/25 | 32.0% | Policy misjudges release timing during memory-guided navigation; memory-to-reality offsets accumulate for narrow receptacles. |
| Head–Arm Coordination Mismatch | 5/25 | 20.0% | Head orients toward one arm's workspace while the other lacks visual guidance, causing failed handovers in dual-arm tasks (Task 5). |

*Table 14.* Failure mode analysis on real-world OOV tasks (25 sampled failed episodes, 5 per task). Failures predominantly arise when translating reliable spatial localization into accurate motor execution under dynamic viewpoints.

| Failure Mode | Count | Prop. | Description |
|---|---|---|---|
| Noisy Spatial Tokens from Imperfect 3D Estimation | 22/50 | 44.0% | VGGT geometry noise on thin/textureless objects propagates through retrieval, misleading action prediction. |
| Irrelevant Memory Activation During Retrieval | 16/50 | 32.0% | Cross-attention activates task-irrelevant memory entries in multi-object scenes, diluting the action signal. |
| Lack of Task-Phase Awareness in Memory | 12/50 | 24.0% | Object-level memory refinement cannot distinguish meaningful state changes (e.g., drawer open vs. closed) from minor visual variations, yielding no phase-transition signal in multi-stage tasks. |

*Table 15.* Failure mode analysis on the fully observable RoboCasa Tabletop GR1 simulation (50 sampled failures, 10 per category). Under full observability, failures reflect limitations of the memory mechanism itself rather than OOV conditions.

*ory Refinement* process severely limits the agent's ability to adapt to scene changes, resulting in the worst performance and confirming that *Dynamic Memory Refinement* is necessary for dynamic environments.

**Processing Intervals.** We analyzes the impact of different frame sampling intervals from *Spatial Memory Construction* as shown in Table 11. Shorter intervals (10) result in redundant and noisy memory updates with a lower average success rate of 44.9%, while excessively large intervals (30/40) lead to sparse scene representations and degraded performance (44.9%/44.4%). An interval of 20 achieves the best balance between information density and temporal diversity, yielding the highest average SR of 49.3%, which confirms that moderate sampling facilitates more stable and informative spatial memory construction. We chose 20 as the sampling interval to obtain the superiority performance.

**Memory Preprocess.** We further assess the cost of each module in the *memory preprocessing stage* outlined in Section 4.2. As shown in Figure 8, removing VGGT (Wang et al., 2025b) (lacking geometric cues) reduces the runtime and GPU cost most significantly but causes a 4.2% performance drop, confirming geometry modeling as the dominant yet essential cost. Excluding DINOv3 (Siméoni et al., 2025) (lacking object semantics) offers a smaller speedup but a larger accuracy loss (-5.6%), showing that semantic encoding is critical for maintaining spatial–semantic consistency despite moderate overhead.

### D.6. Latency Evaluation

Table 12 reports the online inference latency per action chunk for diverse VLA methods in real-world settings. Despite introducing additional parameters for spatial memory modeling and geometry-aware perception, SOMA operates at a comparable inference speed to prior VLA baselines. The measured latency remains within the same order of magnitude as StarVLA (Community, 2026), SpatialVLA (Qu et al., 2025), and GR00T-N1.5 (Bjorck et al., 2025), indicating that the proposed spatial memory design does not impose prohibitive computational overhead during online execution. This allows SOMA to achieve improved manipulation performance without sacrificing practical inference efficiency.

**Geometry Source and Alignment.** Table 13 compares RGB-D based point cloud fusion with VGGT-predicted geometry for spatial memory construction. Although RGB-D fusion leverages depth sensors, its end-to-end latency remains non-trivial due to multi-view point cloud registration and alignment, resulting in comparable preprocessing cost to VGGT-based geometry. More importantly, real-world depth measurements are subject to sensor noise, missing depth, and misalignment across viewpoints, which accumulate during fusion and degrade the geometric consistency of the global memory. This noisy and unstable geometry leads to less reliable spatial tokens and ultimately lower OOV success rates. In contrast, VGGT provides more stable cross-view geometry and pose estimates under active head motion, yielding higher task success despite similar preprocessing latency. These results suggest that, for OOV manipulation, geometric consistency is more critical than raw depth fidelity, motivating our choice of VGGT-based geometry for spatial memory construction.

### D.7. Failure Case Analysis

To better understand the limitations of our approach, we collected 25 representative failed trajectories from real-world OOV tasks (5 per task) and 50 from the fully observable RoboCasa Tabletop GR1 simulation (10 per category), each with full recordings and memory states. As summarized in

Table 14, the dominant real-world failures stem from executing precise manipulation during active visual search: while the spatial memory provides reliable localization, translating it into accurate motor execution under dynamic viewpoints remains a key challenge, accounting for 48.0% of failures via inaccurate grasp pose estimation alone. In the fully observable simulation setting (Table 15), where OOV conditions are absent, failures instead expose limitations of the memory mechanism itself—primarily noisy spatial tokens from imperfect 3D estimation (44.0%) and irrelevant memory activation during retrieval (32.0%). These observations suggest that future improvements should focus on viewpoint-robust grasp estimation and task-phase-aware memory refinement.

### D.8. Limitations and Future Directions

SOMA establishes out-of-vision manipulation under controlled tabletop conditions, and three limitations point to natural directions for future work.

**Robustness in dynamic and parameter-sensitive settings.** Spatial Memory Construction relies on a short-horizon scan in a quasi-static scene with globally fixed thresholds for instance association and similarity-aware fusion. While Dynamic Memory Refinement handles objects that move and re-enter the head camera, fully dynamic scenes, noisier egomotion, and dense clutter would benefit from real-time tracking, confidence-based re-scanning, and per-token geometric confidence; learning these matching criteria, rather than hand-tuning them, is a promising direction. Residual failures further stem less from memory localization than from translating reliable spatial cues into precise motor execution while the head camera is still moving, motivating tighter head–arm coordination and viewpoint-stabilized action decoding.

**Scope, scalability, and semantic granularity.** Memory refinement currently operates at object-level granularity (position and appearance) and cannot distinguish semantically meaningful state transitions—e.g., a drawer open vs. closed—from minor visual variations, limiting performance on multi-stage tasks; augmenting spatial tokens with explicit phase-transition signals is a natural extension. Scaling beyond tabletop settings—to mobile manipulation or room-scale environments—will require hierarchical memory structures operating across spatial and temporal scales, together with loop closure or SLAM-based alignment to control drift, while preserving the core principle of memory-grounded perception and action.

**Safety and societal considerations.** Persistent spatial memory introduces risks that warrant attention in deployment: stale or incorrect memory can compromise physical safety in human-shared environments, and memory-grounded embodied systems raise privacy and potential-misuse concerns. We view confidence scoring, re-scan triggers, memory expiration, on-device storage, and human-in-the-loop confirmation as practical mitigations.

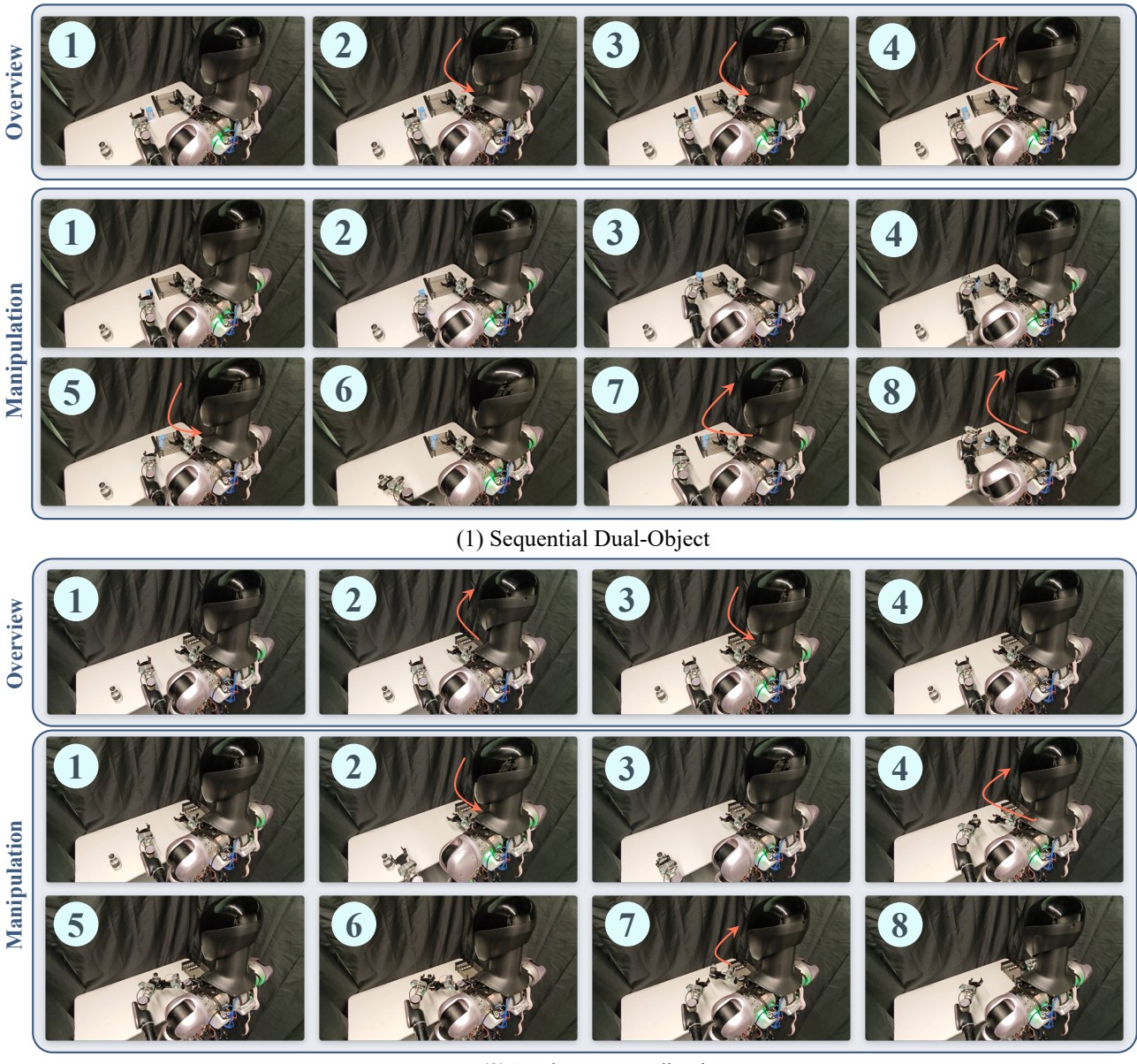

(1) Sequential Dual-Object

(2) Dual-Arm Coordination

*Figure 9.* Illustration of more execution examples including (1) *Sequential Dual-Object* and (2) *Dual-Arm Coordination* using our ***SOMA***.

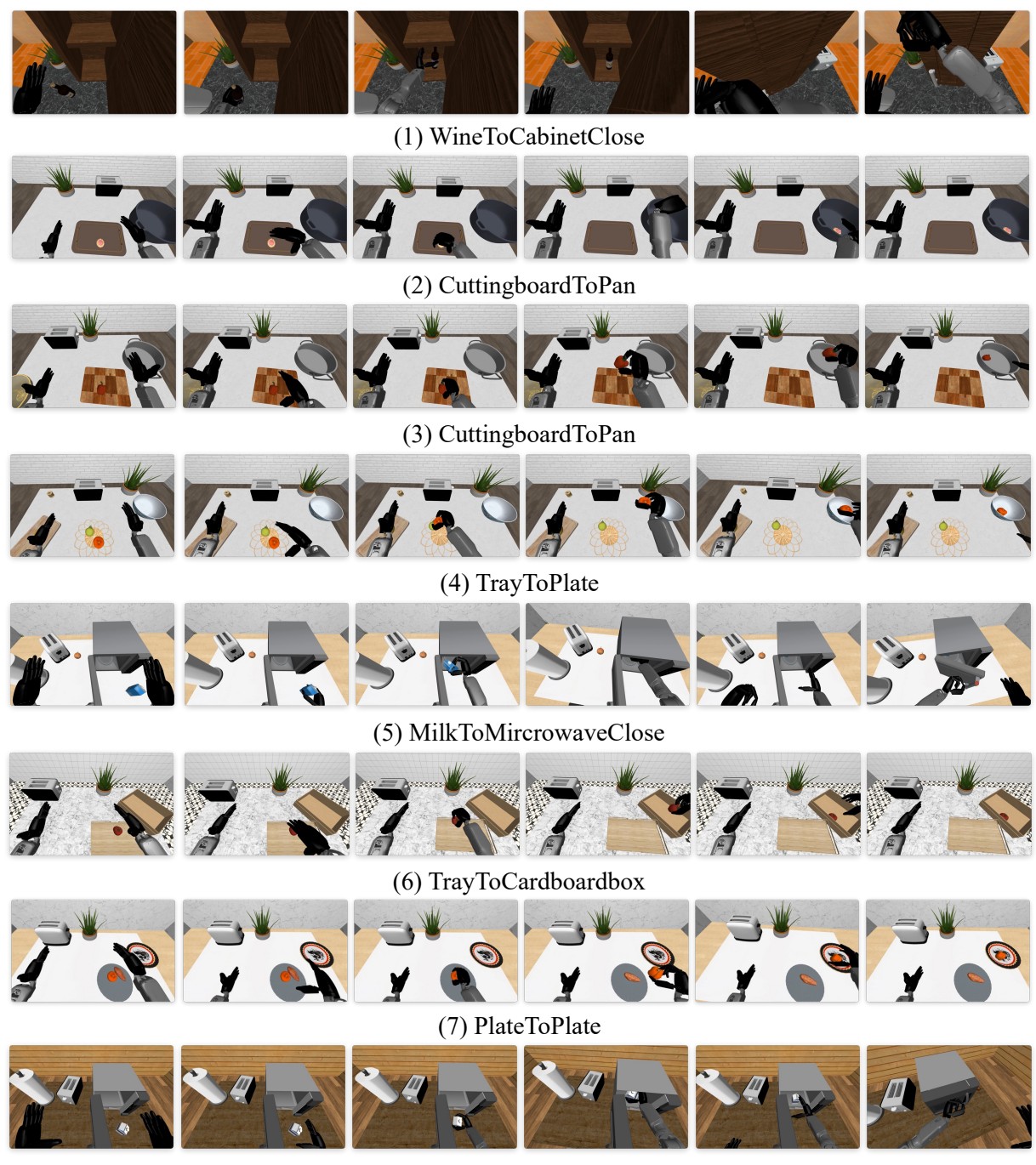

*Figure 10.* Qualitative results of the proposed SOMA framework on the *Robocasa Tabletop GR1* benchmark (Bjorck et al., 2025; Nasiriany et al., 2024). The environment's high-DOF control (arms, hands, waist) generates dynamic and highly variable head camera viewpoints, demonstrating SOMA's robust capability in utilizing its designed spatial memory amidst complex visual instability.

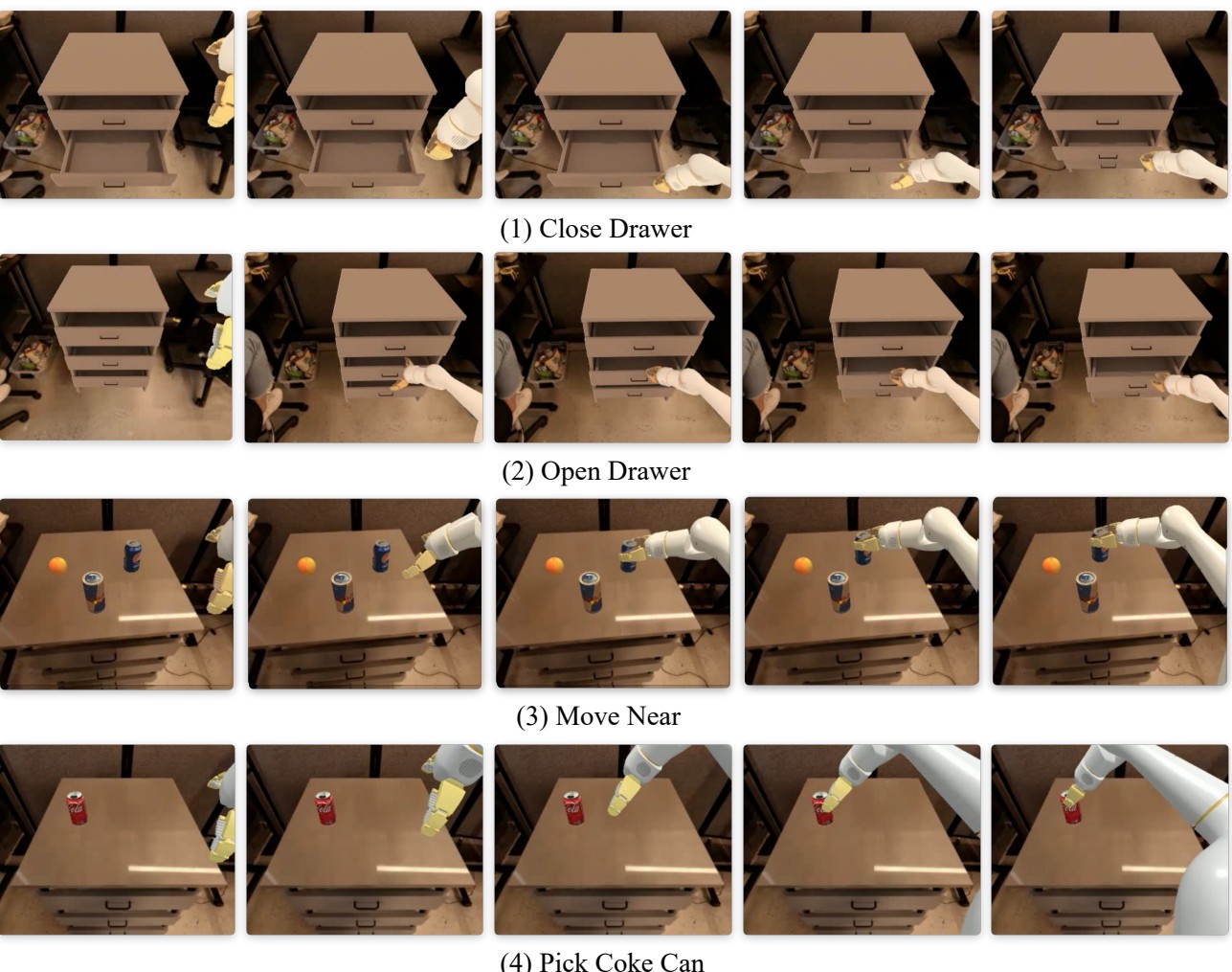

(1) Close Drawer

(2) Open Drawer

(3) Move Near

(4) Pick Coke Can

*Figure 11.* Qualitative Results of SimplerEnv Fractal suite (Li et al., 2024; Zitkovich et al., 2023) using our proposed SOMA.

